# Electrocatalytic valorization of lignocellulose-derived aromatics at industrial-scale current densities

Tao Peng[1,2], Wenbin Zhang[1,2], Baiyao Liang[1], Guanwu Lian[1], Yun Zhang[1] & Wei Zhao [1]✉

Electrocatalytic hydrogenation of lignocellulosic bio-oil to value-added chemicals offers an attractive avenue to use the increasing intermittent renewable electricity and biomass-derived feedstocks. However, to date the partial current densities to target products of these reactions are lower than those needed for industrial-scale productivity, which limits its prospects. Here we report a flow-cell system equipped with a Rh diffusion electrode to hydrogenate lignocellulose-derived aromatic monomers, such as furans and lignin monomers, to value-added chemicals. We achieve high faradaic efficiencies up to 64% at industrial-scale current densities of 300–500 mA cm$^{-2}$, representing high productivities to target products. A screening of electrocatalysts indicates that only by highly-electrolyte-permeable Rh diffusion electrodes are we able to unite current density with faradaic efficiency. We apply in-situ infrared reflection–absorption spectroscopy to investigate the electrode-potential-dependent reaction pathways and intermediates, confirming a wide potential window for efficient electrocatalytic hydrogenation of lignocellulose-derived aromatics to target products.

Lignocellulosic bio-oil[1–3] produced through fast pyrolysis of inedible woody materials[4–6] consists mostly of aromatics (40–55 wt%) (furans and phenolic lignin monomers) (Fig. 1a)[7,8], and thus offers a potential alternative to fossil feedstocks for chemicals production[2,9,10]. Thermocatalytic pyrolysis has been well-developed to produce bio-oil from various sources[4]. However, lignocellulosic bio-oil with reactive aromatics tends to polymerize, known as aging, leading to increased viscosity following storage[11,12]. Reducing the reactive aromatics to more stable and valuable forms thus becomes a preferable biorefinery process.

Efforts have been devoted to developing catalytic systems to upgrade bio-oil aromatics. Currently, thermocatalytic hydrogenation (TCH) is usually used to reduce and stabilize bio-oil aromatics. However, its reaction temperatures (100–500 °C) and hydrogen gas pressure (2–200 bar) add energy-intensity and capital complexity, as well as accelerating undesired polymerization of bio-oil aromatics[13–16].

Moreover, the "gray hydrogen" used in current TCH is produced industrially accompanying substantial CO$_2$ emission.

Electrocatalytic hydrogenation (ECH), which proceeds under mild conditions, can be powered using green electricity produced by renewable energy like solar and wind[17–19], directly utilize proton generated from water to hydrogenate the aromatics of bio-oil to value-added chemicals[20]. These enable ECH to proceed under mild conditions (e.g., ambient temperature and pressure), avoiding utilization of hydrogen gas and severe polymerization of bio-oil. Bio-oil aromatic monomers are readily soluble in water, and thus are favorable feedstock for aqueous electrocatalytic system. However, to date, both faradaic efficiency (FE) and productivity for the ECH of bio-oil aromatics are militated against by competing hydrogen evolution reaction (HER), limiting ECH of bio-oil aromatics at low partial current densities to target products (Table S1)[17,21–31]. To produce industrially relevant quantities of products requires larger current densities

[1]Institute for Advanced Study, Shenzhen University, Shenzhen, Guangdong, China. [2]These authors contributed equally: Tao Peng, Wenbin Zhang. ✉e-mail: weizhao@szu.edu.cn

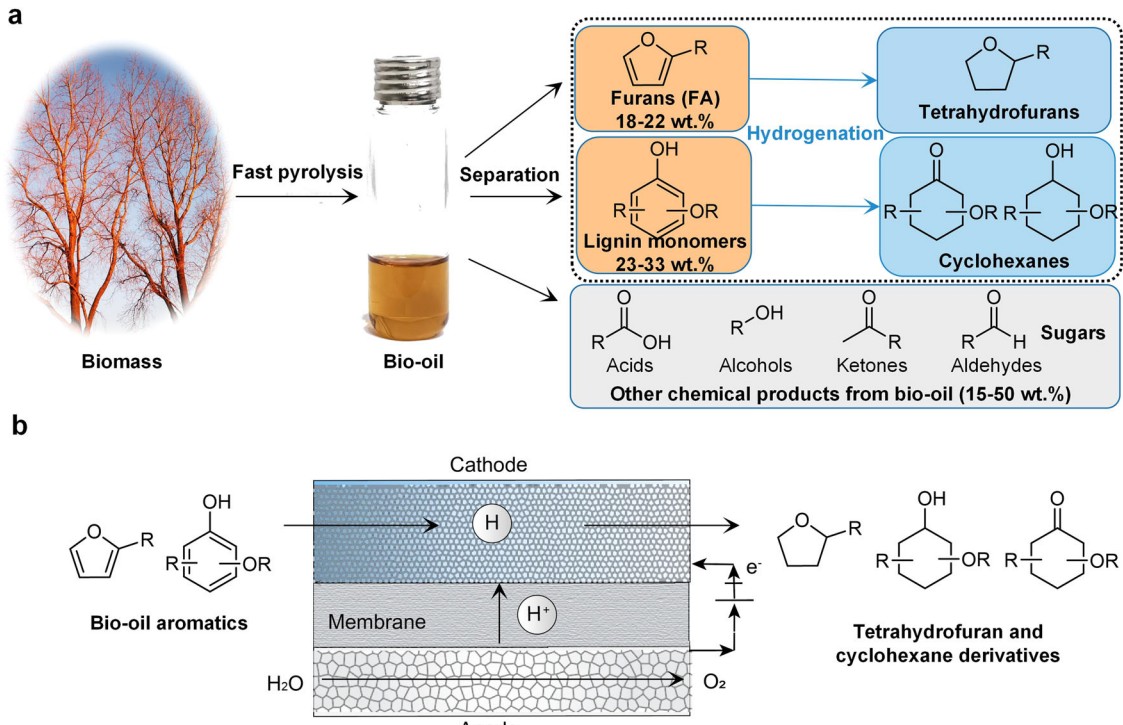

**Fig. 1 | Electrocatalytic hydrogenation of bio-oil aromatics. a** Schematics illustrating lignocellulosic bio-oil and the valorization of bio-oil aromatics (e.g., furans and phenolic lignin monomers) through hydrogenation. Furans (18–22 wt%) and lignin monomers (23–33 wt%) are the dominant contents in bio-oil besides water and ash (25–31 wt% for both). **b** Schematics illustrate the ECH of bio-oil aromatics, i.e., furans and lignin monomers, to more stable and valuable tetrahydrofurans and cyclohexanes derivatives. Rh coated on carbon felt is used as cathode for ECH, IrO$_2$ coated on titanium mesh is used as anode for OER, and Nafion 117 membrane is used for H$^+$ transportation.

(e.g., ~300 mA cm$^{-2}$) with high FEs (e.g., >50%) to achieve higher production efficiency and lower capital costs per unit of products.

Prior ECH studies of bio-oil aromatics have focused on producing bulk chemicals of lower price such as KA oil (the mixture of cyclohexanol and cyclohexanone) ($1600 ton$^{-1}$) from bio-oil derived guaiacol (Figure S1)[29,32], and we find that—compared to production from fossil feedstocks—this has limited prospects for profitability, even if high FE of 80% and current densities of 100–500 mA cm$^{-2}$ were achieved.

We therefore pursued higher-value products such as tetrahydrofurfuryl and methoxy-cyclohexane derivatives (Fig. 1a) from aromatic monomers that can be potentially produced through well-developed thermocatalytic pyrolysis. Tetrahydrofurfuryl derivatives are widely used as solvents (market size of $4.12 billion by 2022) and have a market price range of $3000–$7000 ton$^{-1}$. Methoxy-cyclohexane derivatives ($430 kg$^{-1}$), consisting of -OCH$_3$ on the cyclohexane ring, are used to produce high-value β-lactam antibiotics for human immunodeficiency viruses (HIV)[33].

We sought to employ a flow-cell configuration (Fig. 1b and Figure S2) with our designed membrane electrode assembles (MEA) for ECH of lignocellulosic bio-oil derived aromatics. The cathode layer, proton exchange membrane (PEM) and anode layer are closely packed to fabricate the MEA (Figure S2b and Figure S2c) to shorten the distance between anode and cathode, and this minimizes the Ohmic resistance and cell potentials. A catalyst screening shows that Rh catalysts accelerate ECH of lignin monomer guaiacol and suppress the competing HER compared to other metal catalysts (e.g., Pt, Pd, Cu, Ni, Ir, and Ru) (Figure S3). The Rh nanoparticles of ~5 nm diameter (Figure S4) were coated on the carbon fiber substrate that is common electrolyte diffusion material (Figure S5). This Rh-coated carbon fiber (Rh/CF) diffusion cathode is readily permeable to electrolytes and electrically conductive[34], leading to a high mass transfer rate and a low

full-cell voltage. Using this designed electrocatalytic system, we selectively upgraded three representative lignocellulose-derived aromatic monomers to high value chemicals at industrial-scale productivity level under ambient temperature and pressure.

## Results and discussion
We began examining the electrocatalytic flow-cell system using ECH of 50 mM furfural alcohol (FA)—a model furfuryl compound in bio-oil—to tetrahydrofurfuryl alcohol (THFA) (reaction scheme in Fig. 2a). The flow-cell system shows a significantly lower full-cell voltage of ~3 V at 200 mA cm$^{-2}$ than does the H-cell system (~10 V) (Fig. 2a), owing to lower Ohimc resistance and better mass transport for the flow cell[33]. After a 2-h reaction, the electrolyte temperature slightly increases from 23 °C to 27 °C in a flow-cell system, while the H-Cell has a higher temperature of 32 °C.

We then examined FEs in the current density range of 100–300 mA cm$^{-2}$ using the flow cell. We achieved a high FE of 74% to THFA at a current density of 100 mA cm$^{-2}$ (Fig. 2b). The GC-MS results confirm the production of THFA from ECH of FA (Figure S6). The partial current density is estimated as ~162 mA cm$^{-2}$ to tetrahydrofurfuryl from furans at 300 mA cm$^{-2}$ (FE: 54%), ~6× greater than the best prior reports[21,22] (Table S1). We also investigated the ECH of furfural, and found it produces the mixture of FA and THFA, and FA was eventually converted to THFA with an FE of 41% to THFA (Figure S7).

To evaluate the feasibility of applications of our flow-cell ECH system, we performed a brief technoeconomic analysis (TEA)[35–38] to estimate the plant-gate levelized cost for production of THFA (Table S2, see the details in Supporting Information). The current density strongly affects electricity, capital costs and the associated plant-gate levelized cost of THFA (Fig. 2c). FA ECH at 100 mA cm$^{-2}$ results in a THFA plant-gate levelized cost of $3787 ton$^{-1}$ that exceeds

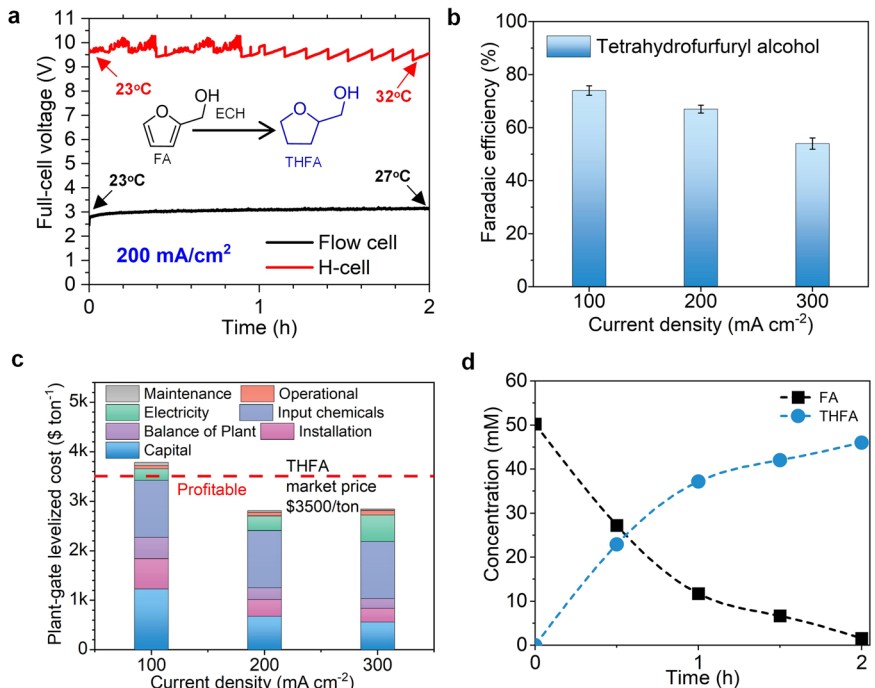

**Fig. 2 | Electrocatalytic hydrogenation of furfural alcohol to tetrahydrofurfuryl alcohol.** **a** Comparison of flow-cell and H-cell at a current density of 200 mA cm$^{-2}$ for 2-h continuous reaction. **b** FE to THFA from FA at current densities of 100–300 mA cm$^{-2}$. Error bars correspond to the standard deviation of three independent measurements. **c** TEA breakdown costs for THFA from FA ECH at various current density. **d** Concentration profile evolution for 2-h reaction at 200 mA cm$^{-2}$ (**b**–**d** flow-cell system).

the market price of $3500 ton$^{-1}$, indicating a limited technoeconomic feasibility in the case of low current density. Increasing the current density to 200 mA cm$^{-2}$ diminishes the needed electrode area and the associated capital cost (electrolyzer, catalyst, and membrane costs), leading to a plant-gate levelized cost lowered down to $2810 ton$^{-1}$ for THFA, i.e., profitable compared to its market price. A further increase to 300 mA cm$^{-2}$ results in a similar plant-gate levelized cost of $2845 ton$^{-1}$, due to the decreased FE (54%) and the increased full-cell voltage (-3.6 V). For 2-h reaction, the flow-cell system shows a 97% conversion rate of FA with a 92% yield of THFA (Fig. 2d).

Lignin monomers are another main ingredient in bio-oil, comprised of syringols and guaiacols—aromatic building blocks with two methoxy groups or one methoxy group on the aryl ring[39]. Compared to guaiacols, the research on the ECH of syringols has to date been limited by its lower kinetics due to an extra methoxy group[23].

We firstly focused on the challenge of ECH of syringol (a typical syringol only consisting of one aryl ring, one hydroxy group, and two methoxy groups), especially at large current densities. The products potentially include multiple cyclohexane derivatives (Fig. 3a): 2-methoxycyclohexanol (2MCHol), 2-methoxycyclohexanone (2MCHN), 2,6-dimethoxycyclohexanol (26MCHol), cyclohexanol (CHol), and cyclohexanone (CHN) (Figure S8). Methoxy-cyclohexanes (2MCHol, 2MCHN, and 26MCHol) are preferred target products. FEs to these desired products (2MCHol, 2MCHN, and 26MCHol) are in the range of 32–52% at ≤ 400 mA cm$^{-2}$ (Fig. 3b). The productivity (a partial current density of -115 mA cm$^{-2}$ at 300 mA cm$^{-2}$) to the target products is 9× greater than the largest-partial-current-density in prior reports (a partial current density of 12.5 mA cm$^{-2}$ using Ru catalysts[24], Table S1). Since 2MCHol dominates the products of ECH of syringol, we estimated the plant-gate levelized cost using 2MCHol. TEA results show plant-gate levelized costs in the range of $40–45 kg$^{-1}$ across 100–400 mA cm$^{-2}$, well below the 2MCHol market price of $430 kg$^{-1}$ (Fig. 3c). We then extended the reaction time to 5-h to estimate the conversion rate and yield to target products at 300 mA cm$^{-2}$ (Fig. 3d). The flow cell shows a roughly constant full-cell voltage -3.1 V in the course of 5-h reaction and 91%

conversion rate with a 64% yield of methoxy-cyclohexanes after 5-h reaction.

We also explored the ECH of guaiacol (a typical guaiacol only consisting of one aryl ring, one hydroxy group, and one methoxy group) to methoxy-cyclohexanes. The products include 2MCHol, 2MCHN, CHol and CHN (Figure S9), and the methoxy-cyclohexanes (2MCHN and 2MCHol) are the desired main products with on Rh/CF catalyst (Fig. 4a). We achieved FEs >44% to methoxy-cyclohexanes across 100–500 mA cm$^{-2}$ (Fig. 4b). The partial current density (-194 mA cm$^{-2}$) to target products at applied current density of 300 mA cm$^{-2}$ is 1.4× greater than the largest-partial-current-density in prior reports (135 mA cm$^{-2}$)[26–28,33]. This partial current density was further increased to 231 mA cm$^{-2}$ at applied 400 mA cm$^{-2}$ with FE of 57.8% to methoxy-cyclohexanes. We continued the ECH of guaiacol at 300 mA cm$^{-2}$ up to 6-h and achieved a conversion rate of 90% with a 59% yield of methoxy-cyclohexanes (Fig. 4c). The result indicates a plant-gate levelized cost of ≤ $9.5 kg$^{-1}$ in the current density range of 100–500 mA cm$^{-2}$, well below the market price of $430 kg$^{-1}$ (Fig. 4d).

We further explored operating stability using 1 liter of catholyte and guaiacol ECH (Figure S10). The flow-cell system maintains an FE > 56% to the target products at a constant current density of 300 mA cm$^{-2}$ across 32-h continuous operation (Fig. 4e). The cathodic potential of Rh/CF is stable around -0.8 V, indicating an excellent durability of this flow-cell electrochemical system. We also conducted cyclic stability experiments in a course of 5 cycles for a total of 60 h (Figure S11). SEM and HRTEM images of the Rh/CF catalyst show negligible morphological change after the stability test (Figure S12a–d). STEM images with EDX mapping results (Figure S12e–g) show that Rh catalysts maintain uniform distribution on carbon fiber, and XPS results (Figure S13) indicate Rh/CF catalysts keep chemical structure and compositional features throughout the 32-h operation. These, along with the stable voltage and FE, indicate that the prepared Rh/CF catalysts have an excellent durability.

We then determined the electrochemical surface area (ECSA), specific current density, and turnover frequency (TOF) to understand

the intrinsic electrocatalytic activity of Rh/CF catalysts. Cu underpotential deposition (Cu$_{UPD}$) method[40] (Figure S14) indicates that Rh/CF has an ECSA of 5.07 m$^2$ g$^{-1}$, in good agreement with the value of 5.23 m$^2$ g$^{-1}$ derived from hydrogen underpotential deposition (H$_{UPD}$) method (Figure S15). Thus, the specific current density for the target

products (methoxy-cyclohexanes) is estimated as 0.61 mA cm$^{-2}$ at applied current density of 300 mA cm$^{-2}$, representing a ~5× increase compared to prior report (0.11 mA cm$^{-2}$)[30]. The specific current density reaches to 0.73 mA cm$^{-2}$ when increasing geometric current density to 400 mA cm$^{-2}$. The turnover frequency (TOF) is 2070.6 h$^{-1}$ (or 0.58 s$^{-1}$)

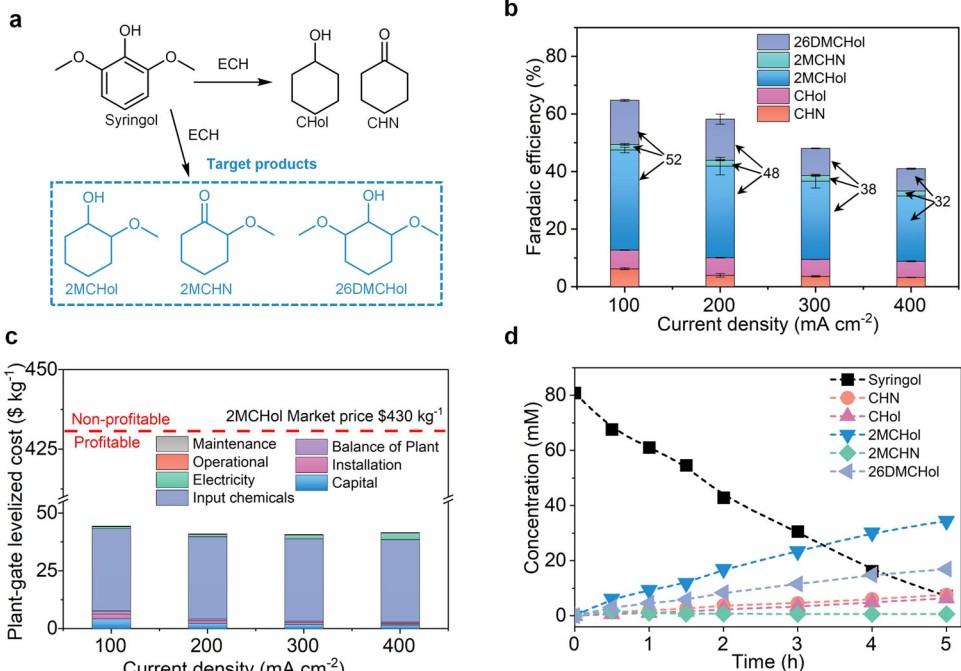

**Fig. 3 | Electrocatalytic hydrogenation of syringol to methoxy-cyclohexanes. a** ECH of syringol to value-added products. **b** FEs to products from syringol at current densities of 100–400 mA cm$^{-2}$. Error bars correspond to the standard deviation of three independent measurements. **c** TEA breakdown costs of high-value methoxy-cyclohexane (2MCHol, the main product) from syringol ECH at different current densities. **d** Concentration profile versus reaction time at 300 mA cm$^{-2}$.

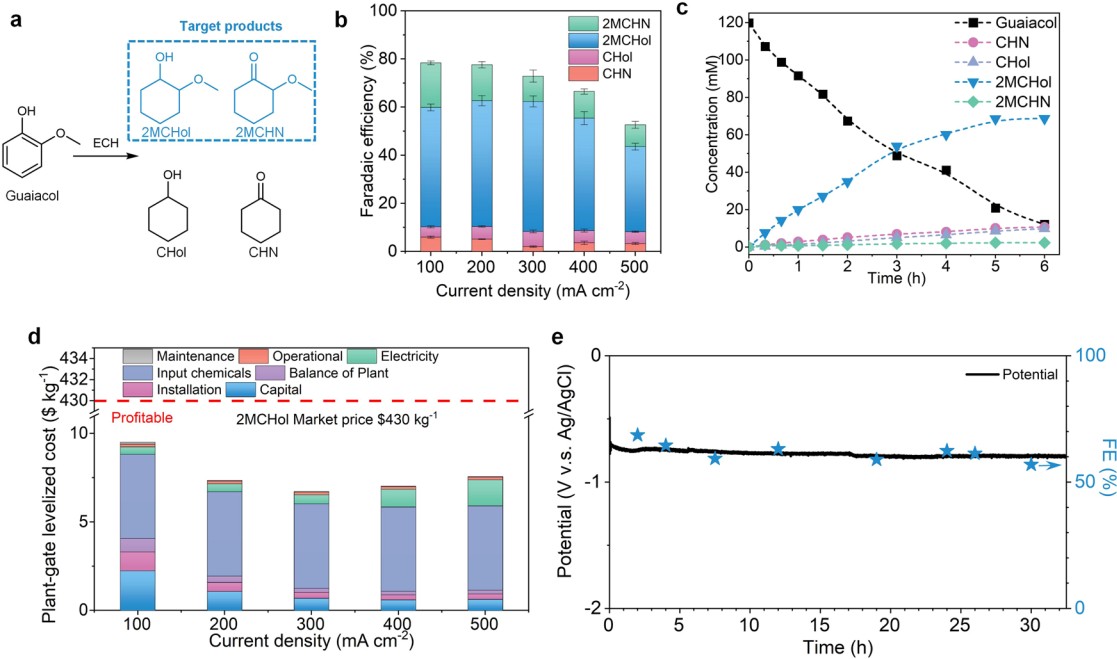

**Fig. 4 | Electrocatalytic hydrogenation of guaiacol ECH to methoxy-cyclohexanes. a** ECH of guaiacol to potential products. **b** FEs to products at current densities of 100–500 mA cm$^{-2}$. Error bars correspond to the standard deviation of three independent measurements. **c** Concentration profile versus time at 300 mA cm$^{-2}$. **d** TEA breakdown costs of high-value methoxy-cyclohexane (2MCHol, the main product) from ECH of guaiacol at different current densities. **e** Evolutions of the cathodic potential and FE to target products for a stable chronopotentiometric operation at 300 mA cm$^{-2}$. The stars mean FE.

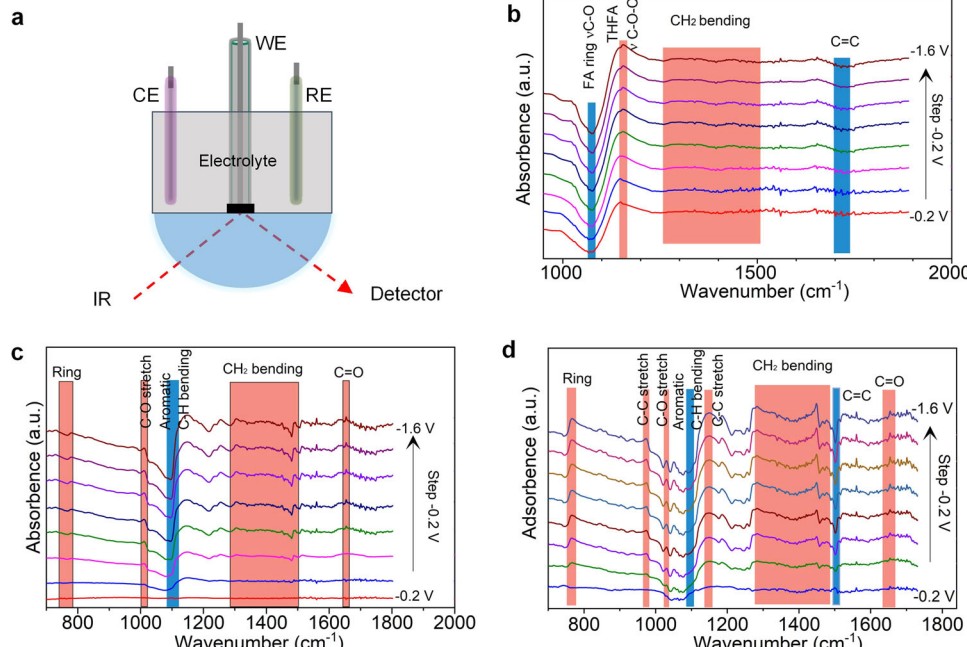

**Fig. 5 | In-situ IRRAS monitoring electrocatalytic hydrogenation of bio-oil aromatics. a** In-situ IRRAS configuration for the ECH of bio-oil aromatics. Rh coated on carbon felt, Ag/AgCl, and Pt wire as the work, reference, and counter electrode, respectively. **b–d** In-situ IRRAS of FA ECH to THFA in 0.2 M HClO$_4$ aqueous solution/30% ethanol mixture electrolyte containing 50 mM FA **b**, syringol ECH in 0.2 M HClO$_4$ aqueous electrolyte containing 80 mM syringol **c**, and guaiacol ECH in 0.2 M HClO$_4$ aqueous electrolyte containing 120 mM guaiacol **d** in a potential range of −0.2 V and −1.6 V vs. Ag/AgCl (with saturated KCl).

for methoxy-cyclohexanes products from ECH of guaiacol at 400 mA cm$^{-2}$, indicating a 3× increase compared to the best value in prior reports[41–43].

Here, we pursued the industrial-scale productivity to the target value-added chemicals. To get an in-depth understanding, we deduced the relationship among the productivity, FE and current density (Supporting Information). We find the productivity is only proportional to FE multiplying total current density, i.e., the partial current density. Therefore, to achieve the industrial-scale productivities, we need to run ECH experiments at large current densities, simultaneously maintain a high FE to target product. However, the "challenge" is achieving a high FE at high current density. Large current densities require high potentials (here more negative), but high potentials lead to stronger competing hydrogen evolution reaction, and cause lowering down of FE. Thus, it is important to balance the current density and FE. We designed our ECH system based on Rh/CF to unite contradictive current density and FE, and achieved industrial-scale productivities at high FEs with a good stability.

We employed electrode-potential-dependent in-situ infrared reflection−absorption spectroscopy (IRRAS) to investigate the ECH reactions (Fig. 5a). Firstly, we monitored the FA ECH reaction by IRRAS (Fig. 5b). We attribute the downward bands centered at 1070 cm$^{-1}$ (C−O vibration in furan ring)[44] and 1700 cm$^{-1}$ (C=C)[45] to the disappearance of FA aromatic ring adsorbed on Rh electrode because of hydrogenation. Upward bands at 1300−1500 cm$^{-1}$ correspond to the appearance of CH$_2$ due to the FA ECH[44]. The growing upward peak at 1150 cm$^{-1}$ (C−O−C vibration of THFA) confirms the appearance of the THFA[46]. The Rh electrode thus likely adsorbs FA molecule and generates adsorbed hydrogen (H*) at applied cathodic potentials[47], and FA is then hydrogenated to THFA by adding hydrogen onto the aromatic ring (Figure S16).

Secondly, we explored the syringol ECH reaction using in-situ IRRAS (Fig. 5c). The aromatic C-H bending band (downward band) at 1100 cm$^{-1}$ confirms the disappearance of syringol and the CH$_2$ bending bands (upward band at 1300−1500 cm$^{-1}$) corroborates the generation

of cyclohexane-based products, the result of ECH. The upward peak located at 1013 cm$^{-1}$ (C-O stretch)[48] indicates methanol, generated by the partial cleavage of -OCH$_3$ from syringol molecules. Furthermore, the upward peak at 1650 cm$^{-2}$ (C=O) suggests the generation of 2MCHN as well as CHN. Syringol is thus electrocatalytically hydrogenated to cyclohexane-based products with partial cleavage of the-OCH$_3$ group.

Lastly, we studied guaiacol ECH reaction using in-situ IRRAS (Fig. 5d). Guaiacol ECH shows similar spectra compared to the ECH of syringol. The downward bands at -1100 cm$^{-1}$ (aromatic C-H bending) and 1503 cm$^{-1}$ (C=C) corresponds to the disappearance of aromatic guaiacol[49]. Several upwards indicate cyclohexane-based products such as 767 cm$^{-1}$ (ring), 970 cm$^{-1}$ (C-C stretch of CHol), 1150 cm$^{-1}$ (cyclohexanes), 1300−1500 cm$^{-1}$ (CH$_2$ bending), and 1650 cm$^{-1}$ (C=O of CHN/2MCHN)[50]. The peak located at 1030 cm$^{-1}$ (C-O stretch of methanol)[48] is attributed to the partial cleavage of -OCH$_3$ from guaiacol, indicating the production of CHol and CHN.

In-situ IRRAS confirms the ECH reaction pathways to THFA and methoxy-cyclohexanes, showing that our Rh/CF electrodes catalyze the ECH reactions of the bio-oil aromatic molecules starting at a very low potential (-0.4 V vs. Ag/AgCl) and overlapping in a wide potential window (-0.4 V to -1.6 V vs. Ag/AgCl). It indicates the feasibility of our flow-cell system with porous Rh/CF catalysts for ECH of lignocellulosic bio-oil aromatic compounds. The onset potential for ECH is less negative than for competing HER, which means ECH of bio-oil aromatics is thermodynamically favored over HER based on our ECH system.

We report a flow-cell system equipped with highly-electrolyte-permeable Rh diffusion cathode for ECH of key bio-oil aromatics (furans and lignin monomers) at industrial-scale current densities. In-situ IRRAS results elaborate the ECH reaction mechanism. TEA studies indicate the ECH of bio-oil aromatics is techno-economically feasible using our flow cell at industrial-scale current densities. This work offers a ECH system to engineer flow-cell-based bio-oil refinery at industrial-level productivity.

## Methods

### Catalyst synthesis

We synthesized Rh electrocatalysts using a process of co-impregnation and calcination. The co-impregnation and calcination method offer excellent scalability to prepare uniform large area electrode. Commercial carbon felt was impregnated with 0.3 mL solution containing 200 mM $RhCl_3$ and dried at 60 °C under air. Next, the impregnated carbon felt was calcinated in a furnace under the Ar atmosphere at a temperature of 400 °C for 2 h. The cathode was further reduced in 0.1 M $Na_2SO_4$ solution by applying a potential of -1 V vs. an Ag/AgCl reference electrode (saturated KCl) for 5 min. The Rh loading is about 6.2 mg $cm^{-2}$. Other control catalysts (Pt, Pd, Ru, Ir) were prepared using the same method with precursor solutions of $H_2PtCl_6$, $PdCl_2$, $RuCl_3$, and $H_2IrCl_6$ for Pt, Pd, and Ru catalysts, respectively. The metal loading was maintained at 6.2 mg $cm^{-2}$, consistent with the Rh catalyst. Before the electrochemical hydrogenation (ECH) experiments, these catalysts underwent a similar preparation process. Nickel foam and copper foam were directly employed as Cu and Ni catalysts, respectively. These materials were thoroughly cleaned with water and ethanol prior to the ECH experiments.

### ECH performance and analysis

The ECH of bio-oil aromatics was performed using a two-chamber flow-cell separated by a Nafion 117 membrane. Cathode and anode are closely against the Nafion membrane to minimize the distance between electrodes and the membrane (Figure S17), and this lowers internal Ohmic resistance and accelerates ion transportation and associated ECH rates[30]. For the control experiments in an H-cell, the electrolyte (30 mL) was magnetically stirred at 1000 RPM. The carbon felts loaded with Rh were restricted to a geometric size of 1 cm × 1 cm as work electrodes with thickness of 3 mm (cathode). A total of 50 mL of 0.2 M $HClO_4$ containing a specific concentration of selected bio-oil aromatic molecules (e.g., 50 mM FA, 80 mM syringol, or 120 mM guaiacol) was used as the catholyte. For ECH of FA, catholyte also contained 30% of ethanol and was purged with argon gas to evacuate the air before ECH. The electrolyte was circulated through the flow cell at 150 ml $min^{-1}$ using peristaltic pumps (Lead Fluid BT100S-1). $IrO_2$ on titanium felt and 0.2 M $HClO_4$ solution were used as anode and anolyte, respectively. Cathodic potentials were measured against an Ag/AgCl reference electrode (saturated KCl). Full-cell voltages were measured against anode. In this study, we did not perform iR-compensation.

## Data availability

The authors declare that all data supporting this study are available within the paper and Supplementary Information files. Source data are provided with this paper.

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

## Acknowledgements

We acknowledge support from the Natural Science Foundation of China (21972096) and the Shenzhen Science and Technology Program (JCYJ20190808150615285).

## Author contributions

T.P. and WB.Z. contributed equally. W.Z. supervised this project. T.P. and WB.Z. carried out experiments. B.L. assisted in GC calibration. T.P., WB.Z. and W.Z. wrote the manuscript. Y.Z. provided help in manuscript writing. G.L. helped edit manuscript. All authors discussed the results and assisted in manuscript preparation.

## Competing interests

The authors declare no competing interests.
