## [Peer Review File · Nature Communications]

Electrocatalytic valorization of lignocellulose-derived aromatics at industrial-scale current densitiesREVIEWER COMMENTS

Reviewer #1 (Remarks to the Author):

The present work discusses the electrochemical hydrogenation of lignin monomers and furan. The following recommendations are made for the revised manuscript:

1. There is previous literature as well have that showed the possibility of ECH of lignin model compounds (e.g. guaiacol) at high current densities (≥ 100 mA/cm²) such as: JApplElectrochem 51, 51-63 (2021), Green Chemistry 24, 7469-7480 (2022), ChemSusChem 13, 629-639 (2020), ACS Sust Chem&Eng 9, 13164-13175 (2021). These papers should be mentioned and reviewed in the context of previous work on high current density ECH.
2. Figure 2a shows much lower cell voltage in the flow cell vs the H-cell. Please explain in more detail. Is it that the inter-electrode gap was much lower in the flow cell? If yes, by how much?
3. When exploring the current density effect on the capital cost and showing that higher current density (e.g. 300 mA/cm²) is better than 100 mA/cm² for lowering the capital cost, was the lower Faradaic efficiency at 300 mA/cm² factored in? (FE: 58% as per Fig 2b?).
4. Do the authors foresee a pathway to carry out ECH directly on lignin? Please offer some comments on this topic.

Reviewer #2 (Remarks to the Author):

This manuscript describes an open flow cell system for evaluating organic chemical hydrogenation and deoxygenation using Rh catalyst on carbon felt. The aim of the study is to reveal the impact that higher current densities have on process economics. ECH focusses on three molecules, guaiacol, syringol, and furfuryl alcohol, all of which have been examined by other authors. Novelty resides in the use of the flow system and the Rh/CF catalyst.

Chemical analysis on the catalyst surface was probed in situ using IRRAS, which is used to observe functional group changes during hydrogenation. GC was used to quantify chemical products, though the detector was not specified in the SI nor was the calibration method described, e.g. external calibration, standard addition, etc.

Economic results were included in the manuscript, while assumptions and design selections were contained in the SI. Input chemicals, e.g. guaiacol, syringol, were found to be the largest contributors to levelized cost, owing to their high values reported as \$5/kg and \$30/kg, respectively. Where we these values obtained? Presumably not pyrolysis bio-oil, even though "bio-oil" appears in the manuscript's title. Besides chemical input costs, capital costs were rather high, likely owing to the small scale of production being only 10 tonnes/day. Their economic model was extracted from two Science articles, one by De Luna et al., and the other by Leow et al. The De Luna article uses CO₂ as a feed, while the Leow article uses olefins as feed. No mention is made of two recent TEA articles that employ ECH for

organics, one by Orella et al. "A general technoeconomic model for evaluating emerging electrolytic processes," in Energy Technology, v. 8(11), and one specifically for pyrolysis bio-oils by Das et al. "Technoeconomic analysis of corn stover conversion by decentralized pyrolysis and electrocatalysis," in Sustainable Energy and Fuels v. 6, p. 2823. As this manuscript reports economics, specifically for bio-oil, it seems some mention of how these results compare should be included.

Overall, the justification of assumptions and design choices in the TEA is very brief to non-existent. This is a major area for improvement that is needed to better understand why certain items lead to their respective cost contributions towards the total cost. Such knowledge will help guide future research by other groups, thus showing impact. A sensitivity analysis, adjusting one parameter while holding the others constant, would also aid in understanding the variables that are most important to optimize and control.

Finally, the time-on-stream results in Figure 4e shows the catalyst stability, which is appreciated by this reviewer, though the results are hardly discussed (less than one sentence). This is a missed opportunity by the authors to elaborate on their catalyst stability.

This reviewer suggests removing the term "bio-oil" from the title if the pyrolysis process was not included in the model used to make the guaiacol, syringol, and 2-furfural used in this analysis. Instead, replace "lignocellulosic bio-oil aromatics" with "guaiacol, syringol, and 2-furfural." Some mention should be made of this facilities rather small scale, which partially explains why the values reported might differ from other groups employing pyrolysis with or without ECH.

Reviewer #3 (Remarks to the Author):

In this work, the authors reported a flow-cell system for electrocatalytic hydrogenation (ECH) of bio-oil aromatics (furans and lignin monomers) to value-added chemicals at industrial-scale current densities using highly-electrolyte-permeable Rh electrodes. In-situ infrared reflection-absorption spectroscopy (IRRAS) is applied to analyze the ECH reaction mechanism and intermediates. TEA studies evaluated the techno-economic feasibility of ECH of bio-oil aromatics. This is a very interesting work and the data supports the research results well, but there still remain some controversies about this work. Therefore, this paper should be reconsidered after revisions.

Comments to the Authors:

1. Please further check the grammar, word use, and punctuation throughout your manuscript.
2. Why choose rhodium nanoparticles as a catalyst? How does it perform relative to other reported catalysts? The authors were suggested lists for comparison.
3. Are there by-products for electrocatalytic hydrogenation of furfural alcohol to tetrahydrofurfuryl alcohol? If so, what are the by-products?
4. What is the cycle stability of this catalyst? The authors are advised to supplement this test and make an analysis.

5. Methods for the detection of these products need to be described in more detail.

Reviewer #4 (Remarks to the Author):

Summary: A Rh/C catalyst and flow electrochemical cell were used to study the electrochemical hydrogenation (ECH) of model bio-oil compounds. The reaction results and techno-economic analysis were reported.

The work is claimed to be highly novel because of the high current densities achieved. The statements in the manuscript suggest that the current densities are generally ~ an order of magnitude better than the literature. This is inaccurate and at times unreasonable. There are other manuscripts that have 100's of mA/cm² currents achieved (some cited and others not) and some of the reactions simply have not been reported by more than one article.

Additional Comments:

How is the current density determined? Is a geometric area used or is the apparent surface area found? This will greatly impact the results of the 3D electrode and comparisons to other electrodes.

It is not clear why furfuryl alcohol was selected as a model compound. Most ECH bio-oil studies start with furfural and then reduce to furfuryl alcohol. Also, the literature the work is compared for current densities is the ECH of furfural.

Pg 3/10 line 69. The authors suggest a "new approach to electrocatalytic hydrogenation of bio-oil aromatics", it is not clear what this approach is or if it is new. Seemingly the primary point of the manuscript is the high productivity of the system, however the cause of this productivity is not clear.

Pg 8/10 line 218. The potential dependency of intermediates does not explain the high productivity and FE.

Pg 8/10 Line 224. The review remains unclear on what the strategy is.

Page 6/10 line 171. An extensive screening of electrocatalysts is mentioned. It would support the work if this was further discussed. The catalysts used and screened are not reported.

- What are the noteworthy results?

Noteworthy results: High productivity to desired products, however this is greatly hindered by the characterization and supporting methods. It is not clear whether this is due to catalyst synthesis, reactor setup, or other.

- Will the work be of significance to the field and related fields? How does it compare to the established literature? If the work is not original, please provide relevant references.

No. Established literature shows similar catalysts, similar reactors, and similar reactions and/or similar current densities. As written, the work as if the literature body does not exist or is not significant. The work suggests that the high currents and system are very novel.

Example articles (not a comprehensive list, see others from the same research groups).

- Andrews, Evan, et al. "Performance of base and noble metals for electrocatalytic hydrogenation of bio-oil-derived oxygenated compounds." *ACS Sustainable Chemistry & Engineering* 8.11 (2020): 4407-4418.

- Li, Zhenglong, et al. "Mild electrocatalytic hydrogenation and hydrodeoxygenation of bio-oil derived phenolic compounds using ruthenium supported on activated carbon cloth." *Green Chemistry* 14.9 (2012): 2540-2549.

- Gonzalez-Garcia, Jose, et al. "Characterization of a carbon felt electrode: structural and physical properties." *Journal of Materials Chemistry* 9.2 (1999): 419-426.

- Chen, Mengyuan, Qingxiang Guo, and Yao Fu. "Electrocatalytic hydrogenation of furfural to furfuryl alcohol using platinum supported on activated carbon fibers." *Electrochimica Acta* 135 (2014): 139-146.

- Wijaya, Y. P.; Smith, K. J.; Kim, C. S.; Gyenge, E. L.: Hydrodeoxygenation of lignin related phenolic monomers in polar organic electrolyte via electrocatalysis in a stirred slurry catalytic reactor. *Green Chemistry* 2022, 24, 7469-7480.

- Wijaya, Y. P.; Grossmann-Neuhaeusler, T.; Dhewangga Putra, R. D.; Smith, K. J.; Kim, C. S.; Gyenge, E. L.: Electrocatalytic hydrogenation of guaiacol in diverse electrolytes using a stirred slurry reactor. *ChemSusChem* 2020, 13, 629-639.

- Does the work support the conclusions and claims, or is additional evidence needed?

The claim of a new approach is unsupported. The reviewer is not clear what the approach is, nor is the methodology clear with supporting evidence. The discussion requires additional evidence of characterizing the system (cathode). For example, the works cited within the manuscript discuss the surface area of the cathodes and the difficulty of current density calculations due to the high surface area of the porous carbon fiber electrodes which is missing from the current manuscript. Techniques such as BET could be used to study surface area, and the thickness of the cathode is required.

- Are there any flaws in the data analysis, interpretation and conclusions? - Do these prohibit publication or require revision?

Potentially. The electrode is 3D and the surface area is not discussed. This would impact the current density reached (If the effective area is in fact higher than used in calculations, due to 3D nature) leading to an over estimation of the electrodes productivity. The productivity of the electrode appears to be the main talking point of the article, and hence relies on cathode characterization.

- Is the methodology sound? Does the work meet the expected standards in your field?

As previously mentioned, surface area determination and current density calculations are required to meet the expected standards.

- Is there enough detail provided in the methods for the work to be reproduced?

No. While a short method to prepare the catalyst was given, the catalyst is described as having "About 6.2 mg/cm²" and the reactor as "a two-chamber flow-cell separated by a Nafion 117 membrane". It is

not clear what was the cause of the high productivity and similar electrocatalysts and reactors have been used in previous reports.

Manuscript ID: NCOMMS-23-02088

Title: “*Electrocatalytic valorization of lignocellulosic bio-oil aromatics at industrial-scale current densities*”

We appreciate the referees' suggestions that motivated us to improve the quality of this work, and have revised the manuscript carefully in the light of the referees' advice. Point-by-point responses to the reviewers' comments are provided. Please find hereafter our answers (in **blue**) to the reviewers' comments (in **black**), and for convenience, the changed parts, and newly added parts for the revised manuscript have been highlighted in a yellow background.

We have changed the title to read "Electrocatalytic valorization of lignocellulose-derived aromatics at industrial-scale current densities".

We appreciate the referees' suggestions that motivated us to improve the quality of this work and the manuscript on the following main aspects:

- 1) **Improved contextualization of the work.** In the light of the referees' recommended references, we now better present the prior reported data and the comparison of relevance works.
- 2) **Intrinsic electrocatalytic activity results.** The manuscript now presents BET, ECSA, specific current density, and turnover frequency results according to referees' suggestions. These indicates our Rh/CF diffusion electrode shows over 2x improvement in intrinsic activity compared to the best prior report.
- 3) **Detailed flow-cell system.** We have now attributed the high performance of the flow-cell system to our custom-made membrane electrode assemblies (MEA). This MEA is close-packed and minimizes Ohmic resistance. The cathode of the MEA consists of a Rh-coated porous carbon felt (electrolyte diffusion electrode), namely Rh/CF diffusion electrode, and this porous electrode is readily permeable to electrolytes and electrically conductive, leading to a high mass transport rate and low full-cell voltage.
- 4) **Detailed results and methods.** In the light of referees' suggestion to detail the

results and methods, we have now included the cycle stability, GC detection with internal standard, the structure of membrane electrode.

REVIEWER COMMENTS

Reviewer #1 (Remarks to the Author):

The present work discusses the electrochemical hydrogenation of lignin monomers and furan. The following recommendations are made for the revised manuscript:

Re: Thank you for the valuable recommendations.

1. There is previous literature as well have that showed the possibility of ECH of lignin model compounds (e.g. guaiacol) at high current densities (≥ 100 mA/cm²) such as: JApplElectrochem 51, 51-63 (2021), Green Chemistry 24, 7469-7480 (2022), ChemSusChem 13, 629-639 (2020), ACS Sust Chem&Eng 9, 13164-13175 (2021). These papers should be mentioned and reviewed in the context of previous work on high current density ECH.

Re: We have now added the relevant references to Table S1 and revised the relevant text in the Introduction section.

2. Figure 2a shows much lower cell voltage in the flow cell vs the H-cell. Please explain in more detail. Is it that the inter-electrode gap was much lower in the flow cell? If yes, by how much?

Re: The distance between the cathode and anode is 6 cm in the H-cell (Figure R1), whereas, in the flow cell, the cathode and anode are in close contact with each other, separated only by a proton exchange membrane with a thickness of 183 μ m (nafion 117). This significantly lowers the electricity resistance in the flow cell, leading to much lower cell voltages.

Figure R1. The H-cell (left) and flow-cell (right) systems used in this study, showing the distance between anode and cathode.

3. When exploring the current density effect on the capital cost and showing that higher current density (e.g. 300 mA/cm²) is better than 100 mA/cm² for lowering the capital cost, was the lower Faradaic efficiency at 300 mA/cm² factored in? (FE: 58% as per Fig 2b?).

Re: The current density, FE and full cell voltage were all factored in the TEA assessment. The current density was increased from 100 mA/cm² to 300 mA/cm² with 3x increasement, whereas the Faraday efficiency decreased only by about 10%. The current density and FE strongly affect the Capital cost (membrane and electrode) and the electricity cost (Table S2).

4. Do the authors foresee a pathway to carry out ECH directly on lignin? Please offer some comments on this topic.

Re: Lignin is usually low soluble in water and usually requires the use of expensive organic solvents and expensive ionic liquids (*Sci. Rep. 7, 5041, (2017)*). Organic solvents have high resistance and associated high cell voltage, and this decreases the energy efficiency and FE. In addition, upgrading the lignin need the C-C scission that is currently quite challenge for electrocatalytic process. These limited the direct ECH on lignin.

The aromatic monomers obtained from bio-oil is readily soluble in water. Aqueous ECH is more economic feasible compared to others with organic solvents or ionic

liquids.

The revised now reads (Page 1):

Bio-oil aromatic monomers are readily soluble in water, and thus are favorable feedstock for aqueous electrocatalytic system.

Reviewer #2 (Remarks to the Author):

This manuscript describes an open flow cell system for evaluating organic chemical hydrogenation and deoxygenation using Rh catalyst on carbon felt. The aim of the study is to reveal the impact that higher current densities have on process economics. ECH focusses on three molecules, guaiacol, syringol, and furfuryl alcohol, all of which have been examined by other authors. Novelty resides in the use of the flow system and the Rh/CF catalyst.

1. Chemical analysis on the catalyst surface was probed in situ using IRRAS, which is used to observe functional group changes during hydrogenation. GC was used to quantify chemical products, though the detector was not specified in the SI nor was the calibration method described, e.g. external calibration, standard addition, etc.

Re: We have now detailed the GC detection in the supporting information. This now reads:

(Page 2 in supporting information) The liquid products were quantified using a GC (Agilent 8890) equipped with a DB-WAX UI column (Agilent). An FID detector was used as detector; 4-propyl-cyclohexanone was used as the internal standard. Calibration was performed using the internal standard method, and a standard curve of the product was prepared for quantitative analysis of the experimental results.

2. Economic results were included in the manuscript, while assumptions and design selections were contained in the SI. Input chemicals, e.g. guaiacol, syringol, were found to be the largest contributors to levelized cost, owing to their high values reported as \$5/kg and \$30/kg, respectively. Where we these values obtained?

Re: Dear reviewer, many thanks for your valuable question. The market prices of input chemicals (e.g. guaiacol, syringol) values come from Alibaba website (the world largest B2B platform). This way was also reported by Jiao's group to search for the market prices of input & output chemicals in TEA analysis (*Nat. Sustainability*, 2021, 4, 911).

3. Presumably not pyrolysis bio-oil, even though “bio-oil” appears in the manuscript’s title. Besides chemical input costs, capital costs were rather high, likely owing to the small scale of production being only 10 tonnes/day.

Re: Thank you for the comment. We used commercial aromatic monomers instead of pyrolysis bio-oil. Our TEA model somehow simplifies the relationship between capital cost and production scale as a linear feature when the current density is fixed, which means varying production scale (e.g., from 10 tonnes/day to 100 tonnes/day) does not change the capital cost per unit of product. The capital cost includes electrolyzer, catalyst, membrane, and electrolyte (please see the TEA model details and Table S2 in Supporting Information), and all these costs per unit of product are determined by the geometric area of electrode. When the current density is fixed, the total capital cost increase linearly with the geometric area of electrode in the case of increase of production scale, however, the capital cost per unit of product keeps unchanged. In the case of fixed production scale, increase of current density will decrease of the geometric area of electrode, leading to lowering the capital cost per unit of product, thus lower down the plant-gate levelized cost of the product. That is why the current density is the key to whether the ECH is profitable and the large current density is required for industrial-level production, which is also the motivation for our paper. However, what you point out is correct for the real manufacturing industry, i.e., increasing the production scale would decrease the plant-gate levelized cost of product. But this would also make TEA model much more complex, which has been considered in our TEA model. Herein, we fixed a small scale of production to investigate the effect geometric current density, FE, full-cell potential and capital cost on the plant-gate levelized costs of target products. The high capital cost is due to the high costs for electrolyzer and catalyst (Rh here), which can be reduced by increasing the current density.

4. Their economic model was extracted from two Science articles, one by De Luna et al., and the other by Leow et al. The De Luna article uses CO₂ as a feed, while the

Leow article uses olefins as feed. No mention is made of two recent TEA articles that employ ECH for organics, one by Orella et al. “A general technoeconomic model for evaluating emerging electrolytic processes,” in *Energy Technology*, v. 8(11), and one specifically for pyrolysis bio-oils by Das et al. “Technoeconomic analysis of corn stover conversion by decentralized pyrolysis and electrocatalysis,” in *Sustainable Energy and Fuels* v. 6, p. 2823. As this manuscript reports economics, specifically for bio-oil, it seems some mention of how these results compare should be included.

Re: We have now cited these two suggested important works in the revised manuscript. Actually, these TEA studies have similar models, and mainly include the costs for electricity, capital, balance of plant, feed (input chemicals), operational (separation), and so on. Only some detailed definitions and calculations are a little different.

It is more suitable for us to employ the TEA model used in those two Science papers after we closely compared the TEA analysis in above literatures, even though the feeds are CO₂ and olefins. As shown in Table S2, the definitions of cost terms in our TEA model are very proper for our ECH study. First of all, we used commercial aromatic monomers (furans and lignin monomers) for ECH reactions instead of pyrolysis bio-oil. The literature of “Technoeconomic analysis of corn stover conversion by decentralized pyrolysis and electrocatalysis” (*Sustainable Energy and Fuels* v. 6, p. 2823) developed the TEA model for corn stover conversion, including the cost for pyrolysis, which is not applicable in our ECH studies. For the literature of “A general technoeconomic model for evaluating emerging electrolytic processes” (*Energy Technology*, v. 8(11)), its TEA model is similar to ours, but the definitions of cost terms in our TEA analysis are richer and more specific. Please see the detailed introduction of our TEA model in Supporting Information.

5. Overall, the justification of assumptions and design choices in the TEA is very brief to non-existent. This is a major area for improvement that is needed to better understand why certain items lead to their respective cost contributions towards the total cost. Such knowledge will help guide future research by other groups, thus showing impact. A

sensitivity analysis, adjusting one parameter while holding the others constant, would also aid in understanding the variables that are most important to optimize and control.

Re: Thanks to referee for this suggestion. For our TEA analysis model, the details have been presented in the “Experimental Section” and “Table S2” in the supporting information. The Table S2 shows the equations needed for the calculations. As detailed answer to question 3, TEA analysis model and TEA cost figures in manuscript (Figure 2c, Figure 3c, Figure 4d), the current density, FE and full-cell potential are the most important parameters for the plant-gate levelized cost of product. However, these three parameters are intrinsically related, which means that adjusting one parameter while holding the others constant is infeasible. For instance, in the ECH experiments, increase current density will lower down FE but increase full-cell potential. Ideally, large current density, high FE and low full-cell potential are beneficial to ECH, which is also the goal of this study. Current density is crucial to the profitability of ECH production, since it directly relates to the geometric area of electrode and corresponding capital cost. FE determines the ECH selectivity and utilization efficiency of energy, and the full-cell potential relates to the electricity cost.

Table S2. Model of techno-economic analysis (TEA) for ECH of bio-oil compounds. Modified model from the previous works.^{4,5}

Breakdown of TEA	Details
Capital cost	Electrolyzer (\$1840/m ²) cost + catalyst cost + membrane cost (5% of the electrolyzer) + electrolyte cost
Installation cost	Lang factor (50%) × (electrolyzer cost + catalyst cost + membrane cost)
Maintenance cost	Maintenance frequency (1/day) × maintenance factor (5%) × (electrolyzer cost + catalyst cost + membrane cost)
Balance of plant	Balance of plant factor (35%) × (electrolyzer cost + catalyst cost + membrane cost)
Operational cost	Product separation cost (40.9 \$/ton of the THFA from furfural alcohol ECH, and 145 \$/ton of the KA oil from guaiacol ECH, 2MC from

	guaiacol ECH, and 2MC from syringol ECH) + other operational cost (10% of the electricity cost)
Electricity cost	Full-cell potential, FE, and electricity price
Input chemical cost	Bio-oil compound cost + Electrolyte cost (HClO ₄ cost + Water cost)

6. Finally, the time-on-stream results in Figure 4e shows the catalyst stability, which is appreciated by this reviewer, though the results are hardly discussed (less than one sentence). This is a missed opportunity by the authors to elaborate on their catalyst stability.

Re: We have now detailed the stability test and included STEM、TEM、SEM and XPS results before and after the stability test. This now reads:

(Page 5) We further explored operating stability using 1 liter of catholyte and guaiacol ECH (Figure S10). The flow-cell system maintains an FE > 56% to the target products at a constant current density of 300 mA/cm² across 32-h continuous operation (Figure 4e). The cathodic potential of Rh/CF is stable around -0.8 V, indicating an excellent durability of this flow-cell electrochemical system. We also conducted cyclic stability experiments in a course of 5 cycles for a total of 60 hours (Figure S11). SEM and HRTEM images of the Rh/CF catalyst show negligible morphological change after the stability test (Figure S12 a-d). STEM images with EDX mapping results (Figure S12 e-g) show that Rh catalysts maintain uniform distribution on carbon fiber, and XPS results (Figure S13) indicate Rh/CF catalysts keep chemical structure and compositional features throughout the 32-h operation. These, along with the stable voltage and FE, indicate that the prepared Rh/CF catalysts have an excellent durability.

7. This reviewer suggests removing the term“bio-oil” from the title if the pyrolysis process was not included in the model used to make the guaiacol, syringol, and 2-fufural used in this analysis. Instead, replace “lignocellulosic bio-oil aromatics” with “guaiacol, syringol, and 2-furfural.” Some mention should be made of this facilities rather small

scale, which partially explains why the values reported might differ from other groups employing pyrolysis with or without ECH.

Re: We have revised the title and introduction. In this study, we electrocatalytically hydrogenated guaiacol, syringol, and 2-furfural. These three chemicals are aromatic monomers. We have now mentioned the ECH conversions at small scales according to the referee's suggestion. This now reads:

(Title in Page 1) Electrocatalytic valorization of lignocellulose-derived aromatics at industrial-scale current densities

(Page 3) Using this designed electrocatalytic system, we selectively upgraded three representative lignocellulose-derived aromatic monomers to high value chemicals at industrial-scale productivity level under ambient temperature and pressure.

Reviewer #3 (Remarks to the Author):

In this work, the authors reported a flow-cell system for electrocatalytic hydrogenation (ECH) of bio-oil aromatics (furans and lignin monomers) to value-added chemicals at industrial-scale current densities using highly-electrolyte-permeable Rh electrodes. In-situ infrared reflection-absorption spectroscopy (IRRAS) is applied to analyze the ECH reaction mechanism and intermediates. TEA studies evaluated the techno-economic feasibility of ECH of bio-oil aromatics. This is a very interesting work and the data supports the research results well, but there still remain some controversies about this work. Therefore, this paper should be reconsidered after revisions.

Comments to the Authors:

1. Please further check the grammar, word use, and punctuation throughout your manuscript.

Re: We have revised the manuscript according to the referees' suggestions.

2. Why choose rhodium nanoparticles as a catalyst? How does it perform relative to other reported catalysts? The authors were suggested lists for comparison.

Re: Thank you for this question. In an initial catalyst screening study, we conducted ECH of guaiacol on popular catalysts that was reported in previous studies. It indicates that the Rh more efficiently catalyzes the ECH of guaiacol compared to other metals. We have now included the catalyst screening and detailed catalyst comparison in Table S1 in the supporting information. This now reads:

(Page 3) A catalyst screen shows that Rh catalysts accelerate ECH of lignin monomer guaiacol and suppress the competing HER compared to other single metal catalysts (e.g., Pt, Pd, Cu, Ni, Ir, and Ru) (Figure S3).

Figure S3. Catalyst screening using the ECH of guaiacol at applied current density of 300 mA/cm² using a H-cell system. The cyclohexane-based products include 2MCHol, 2MCHN, CHol, and CHN. This figure indicates the Rh shows good FE to catalyze the hydrogenation of guaiacol to cyclohexane-based products.

(Page 5 in supporting information) **Table S1.** Selected prior reports for electrocatalytic biorefinery of bio-oil aromatics. Abbreviations: Current density (J); faradaic efficiency (FE); partial current density (J_p); KA oil is the mixture of cyclohexanone ketone and cyclohexanol alcohol.

System	Catalyst	T (°C)	Bio-oil compounds	Products	J mA/c m ²	FE %	J _p mA/cm ²	Reference
Flow cell	3D Rh/CF	25	Furfural alcohol	Tetrahydrofurfuryl alcohol	200	67	134	This work
Flow cell	3D Rh/CF	25	Guaiacol	Methoxy-cyclohexane Pharmaceuticals	300	64	192	This work
Flow cell	PtNiB	60	Guaiacol	KA oils	5	80-90	4-4.5	Adv. Funct. Mater. 2019, 29, 1807651. ⁴
H-cell	Pt/C	50	Furfural	Furfural alcohol	30	85	25.5	Electrochim. Acta. 2014, 135 139–146. ⁵
PEM fuel cell	Pd/C	25	Furfural	Tetrahydrofurfuryl alcohol	15	15	2.3	Green Chem. 2013, 15, 1869-1879. ⁶
H-cell	Raney Ni	75	Mequinol	4-Methoxycyclohexanol	8	26	2.1	Green Chem. 2015, 17, 601-609. ⁷
H-cell	Ru	50	Guaiacol	Methoxy-cyclohexanes	25	19	4.8	Green Chem. 2012, 14,

								2540-2549. ⁸
H-cell	Ru	80	Syringol	Methoxy-cyclohexanes	25	49	12.3	Green Chem. 2012, 14, 2540-2549. ⁸
H-cell	Ru	80	Guaiacol	Cyclohexanol	22	45	9.9	ACS Sustain. Chem. Eng. 2019, 7, 8375-8386. ⁹
H-cell	Pt/C	50	Guaiacol	2-methoxycyclohexanone	109	72	78.5	J. Appl. Electrochem. 2021, 5151-63. ¹⁰
H-cell	Pt/C	60	Guaiacol	Cyclohexanone	182	34.61	63	Green Chem. 2022, 24, 7469-7480. ¹¹
H-cell	Pt/C	50	Guaiacol	2-methoxycyclohexanol	109	82	89.4	ChemSusChem 2020, 13, 629-639. ¹²
H-cell	Pt/C	24	Guaiacol	Cyclohexanol	150	75	112.5	ACS Sustainable Chem. Eng. 2021, 9, 13164-13175. ¹³
H-cell	Ru/ACC	80	Guaiacol	2-methoxycyclohexanol	100	30	30	Green Chem. , 2012, 14, 2540-2549. ⁸
Improved H-cell	Dispersed Rh/CF powders	18	Mequinol	4-Methoxycyclohexanone mixture	-	35	-	J. Catal. , 2016, 344 263-272. ²
Flow cell	Pd	25	Benzaldehyde	Benzyl alcohol	5	60	3	ACS Sustain. Chem. Eng. 2020, 8, 4407-4418. ¹⁴

3. Are there by-products for electrocatalytic hydrogenation of furfural alcohol to tetrahydrofurfuryl alcohol? If so, what are the by-products?

Re: The GC-MS results indicate the production of tetrahydrofurfuryl alcohol (THFA) from the ECH of furfural alcohol without other byproducts. We have now included the GC-MS results in supporting information.

4. What is the cycle stability of this catalyst? The authors are advised to supplement this test and make an analysis.

Re: We have now included the cyclic stability test for guaiacol ECH in Supporting Information (Figure S11) according to the referee's suggestion. The flow-cell system with Rh/CF catalysts shows good durability in a course of 5 cycle stability test (12 h for each).

Figure S11. Cyclic stability test of guaiacol ECH on Rh/CF catalyst using flow-cell system at 100 mA/cm². The cathodic potential is stable around -0.3 V and the FE of each cycle is stable at around 60% in a course of 5 cycles for a total of 60 hours.

5. Methods for the detection of these products need to be described in more detail.

Re: We have now added details of GC detection in the supporting information. It now reads:

(Page 2 in SI) The liquid products were collected at one hour and other specific times as indicated in the main text. The liquid samples were qualitatively analyzed using a GC-MS (PerkinElmer Clarus 680) equipped with a Stabilwax column (fused silica, Restek). The liquid products were quantified using a GC (Agilent 8890) equipped with a DB-WAX UI column (Agilent). An FID detector was used as detector; 4-propylcyclohexanone was used as the internal standard. Calibration was performed using the internal standard method, and a standard curve of the product was prepared for quantitative analysis of the experimental results. Injector temperature was set to 250°C with a split ratio of 30:1. The oven program started at 80°C for 1 min, heated to 140°C

at 60°C/min and then to 190°C at 50°C/min, and then to 210°C at 40°C/min and the temperature was held for more than 1 min.

Reviewer #4 (Remarks to the Author):

Summary: A Rh/C catalyst and flow electrochemical cell were used to study the electrochemical hydrogenation (ECH) of model bio-oil compounds. The reaction results and techno-economic analysis were reported.

The work is claimed to be highly novel because of the high current densities achieved. The statements in the manuscript suggest that the current densities are generally ~ an order of magnitude better than the literature. This is inaccurate and at times unreasonable. There are other manuscripts that have 100's of mA/cm² currents achieved (some cited and others not) and some of the reactions simply have not been reported by more than one article.

Re: Thank you so much for your suggestions. We have now revised manuscripts and included suggested references according to referee's suggestions. Please see the responses below.

Additional Comments:

1. How is the current density determined? Is a geometric area used or is the apparent surface area found? This will greatly impact the results of the 3D electrode and comparisons to other electrodes.

Re: The use of geometric current density in this study. In our original text, we determined the current density based on geometric area. Because the current density is usually calculated using geometric area of membrane electrode assemble (MEA) for fuel cell or electrolysis cell (*Nat. Nanotechnol.* 16, 140-147 (2021)). The geometric area of MEA directly affects the total cost of a fuel cell stack (e.g., proton exchange membrane, diffusion layer, catalyst usage, and associated bipolar plate). Increasing geometric current density would lower the total cost of fuel cell stack. In addition, the current densities are usually given by the geometric area of electrode in the electrocatalysis papers.

The TOF and specific current density based on ECSA. The intrinsic activity has been useful in the stage of catalyst design. We have now also included turnover frequency (TOF) and intrinsic current density results based on electrochemical surface area (ECSA) (See the Table R1). The Rh/CF catalysts in the designed flow cell shows 2x-3x increases in intrinsic current density and TOF compared to prior reports.

In detail, the Rh/CF catalysts show a BET specific surface area (SSA) of 12.4 m²/g and ECSA of 8.59 m²/g. The Rh loading is 6.2 mg. We then convert the geometric current density (J= 300 mA/cm²) to intrinsic current density (J_{ECSA} = 0.564 mA/cm²). The partial intrinsic current density is 0.365 mA/cm² to the target products (methoxy-cyclohexanes) from ECH of guaiacol, representing ~3x increase compared to other guaiacol ECH reports. The TOFs are 1158 h⁻¹ and 1402 h⁻¹ on Rh/CF catalyst at applied current density of 300 and 400 mA/cm² respectively, representing a ~2x increase compared to the best prior report (*J. Chem. Phys.* 156, 104703 (2022)).

Table R1. Geometric current density, FE, partial current density, partial intrinsic current density and TOF for guaiacol ECH. ECSA was determined by Cu_{UPD} stripping method. (*ChemElectroChem* 6, 1990-1995 (2019).)

J (mA/cm ²)	FE _{methoxycyclohexanes} (%)	J _p (mA/cm ²)	J _{ECSA} to targets (mA/cm ²)	TOF (h ⁻¹)
300	64.5	194	0.365	1158
400	57.8	231	0.434	1402

This now reads:

(Page 6) We then determined the electrochemically surface area (ECSA), specific current density, and turnover frequency (TOF) to understand the intrinsic electrocatalytic activity of Rh/CF catalysts. BET result indicates a specific surface area (SSA) of 12.2 m²/g for Rh/CF catalysts. The CV results through Cu_{UPD} stripping method⁴¹ (Figure S14) indicate that Rh/CF has an ECSA of 8.59 m²/g. Thus, the specific current density for the target products (methoxy-cyclohexanes) is estimated as 0.37 mA/cm² at applied current density of 300 mA/cm², representing a 3x increase compared to prior reports (0.11 mA/cm²).³⁰ The specific current density reaches to 0.43 mA/cm² when increasing geometric current density to 400 mA/cm². The turnover

frequency (TOF) is 1402 h^{-1} (or 0.39 s^{-1}) for methoxy-cyclohexanes products from ECH of guaiacol at 400 mA/cm^2 , indicating a 2x increase compared to the best value in prior reports.⁴²⁻⁴⁴

2. It is not clear why furfuryl alcohol was selected as a model compound. Most ECH bio-oil studies start with furfural and then reduce to furfuryl alcohol. Also, the literature the work is compared for current densities is the ECH of furfural.

Re: We have now included a preliminary ECH of furfuryl (FF) in the supporting information. The total FE of ECH of FF to FA and THFA is over 50% at 200 mA/cm^2 in a course of 2-h ECH. This now reads:

(Page 4) We also investigated the ECH of furfural, and found it produces the mixture of FA and THFA, and FA was eventually converted to THFA with an FE of 41% to THFA (Figure S7).

Figure S7. GC-MS results of FA and THFA from ECH of FF on Rh/CF catalyst at 200 mA/cm^2 after a course of 1-h reaction. The FF was hydrogenated to FA and THFA with total FE of 50% in a course of 1-h ECH. The FA was eventually converted to THFA with FE of 41% in a course of 2-h ECH. The FE to FA from FF is 14% in a course of 2-h ECH. The FA peaks was determined in Figure S6.

3. Pg 3/10 line 69. The authors suggest a “new approach to electrocatalytic hydrogenation of bio-oil aromatics”, it is not clear what this approach is or if it is new. Seemingly the primary point of the manuscript is the high productivity of the system, however the cause of this productivity is not clear.

Re: The flow-cell with membrane electrode assemblies (MEA). We have improved a flow cell with our custom-made MEA, consisting Rh/CF diffusion electrode, proton exchange membrane (PEM) and anode layer. These three-layer materials are closely packed to shorten the distance between anode and cathode, and this minimizes the Ohmic resistance (as shown in Figure S2) (*J. Catal.* 344, 263-272, (2016)).

The CF diffusion layer was used as electrode substrate of MEA. The Rh catalysts was coated onto a porous carbon felt that is electrolyte diffusion layer. Although this Rh/CF has a limited SSA of 12.2 m²/g (detailed in question 1), it offers an open porous network to electrolyte diffusion and thus improves the mass transfer.

The choice of ECH catalyst. A catalyst screen indicates Rh shows the best performance on ECH aromatics and suppressed competing HER compared to other tested catalysts (e.g., Pt, Pd, Ru, Ni).

The revised now reads:

(Page 3) We sought to employ a flow-cell configuration (**Figure 1b and Figure S2a**) with our designed membrane electrode assemblies (MEA) for ECH of lignocellulosic bio-oil derived aromatics. **The cathode layer, proton exchange membrane (PEM) and anode layer are closely packed to fabricate the MEA (Figure S2b)** to shorten the distance between anode and cathode, and this **minimizes the Ohmic resistance and cell potentials.** A catalyst screening shows that Rh catalysts accelerate ECH of lignin monomer guaiacol and suppress the competing HER compared to other metal catalysts (e.g., Pt, Pd, Cu, Ni, Ir, and Ru) (**Figure S3**). The Rh nanoparticles of ~5 nm diameter (**Figure S4**) were coated on the carbon fiber substrate **that is common electrolyte diffusion material (Figure S5)**. This Rh coated carbon fiber (Rh/CF) diffusion cathode is readily permeable to electrolytes and electrically conductive,³⁵ leading to a high mass transfer rate and a low full-cell voltage. **Using this designed electrocatalytic system, we selectively upgraded three representative lignocellulose-derived aromatic monomers to**

high value chemicals at industrial-scale productivity level under ambient temperature and pressure.

Figure S3. Catalyst screening using the ECH of guaiacol at applied current density of 300 mA/cm² using a H-cell system. The cyclohexane-based products include 2MCHol, 2MCHN, CHol, and CHN. This figure indicates the Rh shows good FE to catalyze the hydrogenation of guaiacol to cyclohexane-based products.

4. Pg 8/10 line 218. The potential dependency of intermediates does not explain the high productivity and FE.

Re: The potential dependency of intermediates implies the ECH reactions have high selectivity to target products, and preferentially occur over the competing HER, leading to high FEs.

To elaborate the origin of high productivity, we have now included the ECSA and intrinsic activity (e.g., specific current density and TOF), indicating the intrinsic activity of the Rh/CF catalyst is excellent.

Thank you so much for this comment, we have deleted this expression in order to be rigorous.

5. Pg 8/10 Line 224. The review remains unclear on what the strategy is.

Re: Detailed in question 3

6. Page 6/10 line 171. An extensive screening of electrocatalysts is mentioned. It would support the work if this was further discussed. The catalysts used and screened are not reported.

Re: We have now added the catalyst screening results. This now reads:

(Page 3) A catalyst screening shows that Rh catalysts accelerate ECH of lignin monomer guaiacol and suppress the competing HER compared to other metal catalysts (e.g., Pt, Pd, Cu, Ni, Ir, and Ru) (Figure S3).

Figure S3. Catalyst screening using the ECH of guaiacol at applied current density of 300 mA/cm² using a H-cell system. The cyclohexane-based products include 2MCHol, 2MCHN, CHol, and CHN. This figure indicates the Rh shows good FE to catalyze the hydrogenation of guaiacol to cyclohexane-based products

7. - What are the noteworthy results?

Noteworthy results: High productivity to desired products, however this is greatly hindered by the characterization and supporting methods. It is not clear whether this is due to catalyst synthesis, reactor setup, or other.

Re: We have now detailed our discussion and methods for flow-cell system. This now reads:

(Page 3) We sought to employ a flow-cell configuration (Figure 1b and Figure S2)

with our designed membrane electrode assemblies (MEA) for ECH of lignocellulosic bio-oil derived aromatics. The cathode layer, proton exchange membrane (PEM) and anode layer are closely packed to fabricate the MEA (Figure S2b) to shorten the distance between anode and cathode, and this minimizes the Ohmic resistance and cell potentials. A catalyst screening shows that Rh catalysts accelerate ECH of lignin monomer guaiacol and suppress the competing HER compared to other metal catalysts (e.g., Pt, Pd, Cu, Ni, Ir, and Ru) (Figure S3). The Rh nanoparticles of ~5 nm diameter (Figure S4) were coated on the carbon fiber substrate that is common electrolyte diffusion material (Figure S5). This Rh coated carbon fiber (Rh/CF) diffusion cathode is readily permeable to electrolytes and electrically conductive,³⁵ leading to a high mass transfer rate and a low full-cell voltage. Using this designed electrocatalytic system, we selectively upgraded three representative lignocellulose-derived aromatic monomers to high value chemicals at industrial-scale productivity level under ambient temperature and pressure.

(Page 2 in SI) **Electrocatalytic hydrogenation performance and product analysis.**

The ECH of bio-oil compounds was performed using a two-chamber flow-cell separated by a Nafion 117 membrane. Cathode and anode are closely against the Nafion membrane to minimize the distance between electrodes and the membrane (Figure S2b), and this lowers internal Ohmic resistance and accelerates ion transportation and associated ECH rates².

Figure S2. (a) The electrolyte was circulated through the flow cell. (b) Schematic illustrating our custom-made membrane electrode assemblies (MEA) consisting of a catalyst coated diffusion layer, proton exchange membrane (PEM) and IrO₂/Ti mesh anode. These three layers are closely packed to minimize the Ohmic resistance and electrode potentials.

8. - Will the work be of significance to the field and related fields? How does it compare to the established literature? If the work is not original, please provide relevant references.

No. Established literature shows similar catalysts, similar reactors, and similar reactions and/or similar current densities. As written, the work as if the literature body does not exist or is not significant. The work suggests that the high currents and system are very novel.

Example articles (not a comprehensive list, see others from the same research groups).

- Andrews, Evan, et al. "Performance of base and noble metals for electrocatalytic hydrogenation of bio-oil-derived oxygenated compounds." *ACS Sustainable Chemistry & Engineering* 8.11 (2020): 4407-4418.
- Li, Zhenglong, et al. "Mild electrocatalytic hydrogenation and hydrodeoxygenation of bio-oil derived phenolic compounds using ruthenium supported on activated carbon cloth." *Green Chemistry* 14.9 (2012): 2540-2549.
- Gonzalez-Garcia, Jose, et al. "Characterization of a carbon felt electrode: structural and physical properties." *Journal of Materials Chemistry* 9.2 (1999): 419-426.
- Chen, Mengyuan, Qingxiang Guo, and Yao Fu. "Electrocatalytic hydrogenation of furfural to furfuryl alcohol using platinum supported on activated carbon fibers." *Electrochimica Acta* 135 (2014): 139-146.
- Wijaya, Y. P.; Smith, K. J.; Kim, C. S.; Gyenge, E. L.: Hydrodeoxygenation of lignin related phenolic monomers in polar organic electrolyte via electrocatalysis in a stirred slurry catalytic reactor. *Green Chemistry* 2022, 24, 7469-7480.
- Wijaya, Y. P.; Grossmann-Neuhaeusler, T.; Dhewangga Putra, R. D.; Smith, K. J.; Kim, C. S.; Gyenge, E. L.: Electrocatalytic hydrogenation of guaiacol in diverse electrolytes using a stirred

slurry reactor. ChemSusChem 2020, 13, 629-639.

Re: We have now added the suggested literatures into Table S1 to compare the activity and selectivity of the catalysts. Our flow-cell system with Rh/CF catalysts still shows better intrinsic activity and partial current density. The revised introduction and SI now read:

However, to date, both faradaic efficiency (FE) and productivity for the ECH of bio-oil aromatics are militated against by competing hydrogen evolution reaction, limiting ECH of bio-oil aromatics at low partial current densities to target products (Table S1).^{17,21-31} To produce industrially relevant quantities of products requires larger current densities (e.g., ~300 mA/cm²) with high FEs (e.g., >50%) to achieve higher production efficiency and lower capital costs per unit of productivity.

Table S1. Selected prior reports for electrocatalytic biorefinery of bio-oil aromatics. Abbreviations: Current density (J); faradaic efficiency (FE); partial current density (J_p); KA oil is the mixture of cyclohexanone ketone and cyclohexanol alcohol.

System	Catalyst	T (°C)	Bio-oil compounds	Products	J mA/cm ²	FE %	J _p mA/cm ²	Reference
Flow cell	3D Rh/CF	25	Furfural alcohol	Tetrahydrofurfuryl alcohol	200	67	134	This work
Flow cell	3D Rh/CF	25	Guaiacol	Methoxy-cyclohexane Pharmaceuticals	300	64	192	This work
Flow cell	PtNiB	60	Guaiacol	KA oils	5	80-90	4-4.5	Adv. Funct. Mater. 2019, 29, 1807651. ⁴
H-cell	Pt/C	50	Furfural	Furfural alcohol	30	85	25.5	Electrochim. Acta. 2014, 135 139–146. ⁵
PEM fuel cell	Pd/C	25	Furfural	Tetrahydrofurfuryl alcohol	15	15	2.3	Green Chem. 2013, 15, 1869-1879. ⁶
H-cell	Raney Ni	75	Mequinol	4-Methoxycyclohexanol	8	26	2.1	Green Chem. 2015, 17, 601-609. ⁷
H-cell	Ru	50	Guaiacol	Methoxy-cyclohexanes	25	19	4.8	Green Chem. 2012, 14, 2540-2549. ⁸
H-cell	Ru	80	Syringol	Methoxy-cyclohexanes	25	49	12.3	Green Chem. 2012, 14, 2540-2549. ⁸
H-cell	Ru	80	Guaiacol	Cyclohexanol	22	45	9.9	ACS Sustain. Chem. Eng. 2019, 7, 8375-8386. ⁹

H-cell	Pt/C	50	Guaiacol	2-methoxycyclohexanone	109	72	78.5	J. Appl. Electrochem. 2021, 5151-63. ¹⁰
H-cell	Pt/C	60	Guaiacol	Cyclohexanone	182	34.61	63	Green Chem. 2022, 24, 7469-7480. ¹¹
H-cell	Pt/C	50	Guaiacol	2-methoxycyclohexanol	109	82	89.4	ChemSusChem 2020, 13, 629-639. ¹²
H-cell	Pt/C	24	Guaiacol	Cyclohexanol	150	75	112.5	ACS Sustainable Chem. Eng. 2021, 9, 13164-13175. ¹³
H-cell	Ru/ACC	80	Guaiacol	2-methoxycyclohexanol	100	30	30	Green Chem. , 2012, 14, 2540-2549. ⁸
Improved H-cell	Dispersed Rh/CF powders	18	Mequinol	4-Methoxycyclohexanone mixture	-	35	-	J. Catal. , 2016, 344 263-272. ²
Flow cell	Pd	25	Benzaldehyde	Benzyl alcohol	5	60	3	ACS Sustain. Chem. Eng. 2020, 8, 4407-4418. ¹⁴

9. - Does the work support the conclusions and claims, or is additional evidence needed?

The claim of a new approach is unsupported. The reviewer is not clear what the approach is, nor is the methodology clear with supporting evidence. The discussion requires additional evidence of characterizing the system (cathode). For example, the works cited within the manuscript discuss the surface area of the cathodes and the difficulty of current density calculations due to the high surface area of the porous carbon fiber electrodes which is missing from the current manuscript. Techniques such as BET could be used to study surface area, and the thickness of the cathode is required.

Re: The good performance is due to the improved flow cell with our custom-made MEA, consisting Rh/CF diffusion electrode, proton exchange membrane (PEM) and anode layer. (Detailed in question 3)

We now have added BET, ECSA results and further determined the specific current density and TOF. (Detailed in question 1)

The thickness of carbon felt is 3 mm. We checked some other organic electrosynthesis literature, and found that the thickness of carbon felts previously used was in the range of 3-6.35 mm (*ACS Sustainable Chem. Eng.* 2020, 8, 4407-4418; *ACS*

Catal. 2019, 9, 11, 9964–9972; *ACS Catal.* 2022, 12, 19, 11910–11917; *J. Am. Chem. Soc.* 2021, 143, 41, 17226–17235; *ACS Sustainable Chem. Eng.* 2019, 7, 13, 11138–11149).

The BET results shows that the Rh/CF has about 12.2 m²/g. This is comparable to the previous reported BET and SSA of CF (8 m²/g, *J. Mater. Chem.* 9, 419-426 (1999)). The revised method now reads:

(Page 2 in supporting information) For the control experiments in an H-cell, the electrolyte (30 mL) was magnetically stirred at 1000 RPM. The carbon felts loaded with Rh were restricted to a geometric size of 1 cm x 1 cm as work electrodes with thickness of 3 mm (cathode).

10. - Are there any flaws in the data analysis, interpretation and conclusions? - Do these prohibit publication or require revision?

Potentially. The electrode is 3D and the surface area is not discussed. This would impact the current density reached (If the effective area is in fact higher than used in calculations, due to 3D nature) leading to an over estimation of the electrodes productivity. The productivity of the electrode appears to be the main talking point of the article, and hence relies on cathode characterization.

Re: We thank the reviewer for the suggestions. We have now included ECSA, specific current density, and TOF for guaiacol ECH on Rh/CF catalysts. (See the response to the question 1).

11- Is the methodology sound? Does the work meet the expected standards in your field? As previously mentioned, surface area determination and current density calculations are required to meet the expected standards.

Re: Thank you for the suggestions. We now included the method for measuring ECSA, geometric current density, and specific current density. These were detailed in question 1 (See the response to the Question 1).

12.- Is there enough detail provided in the methods for the work to be reproduced?

No. While a short method to prepare the catalyst was given, the catalyst is described as having “About 6.2 mg/cm²” and the reactor as “a two-chamber flow-cell separated by a Nafion 117 membrane”. It is not clear what was the cause of the high productivity and similar electrocatalysts and reactors have been used in previous reports.

Re: We have now detailed our method for flow-cell set-up. The high efficiency of this system is mainly due to two reasons:

1) the good intrinsic activity of the Rh catalyst (addressed in question 1);

2) the flow-cell system with an MEA that shorten the distance of anode and cathode, and this increases the mass transfer, lowers the cathodic potential, and suppresses the competing HER. These together shift the reaction selectivity to our target products *via* ECH reactions.

The revised the methods of catalyst and flow-cell preparation now read:

(Page 3) We sought to employ a flow-cell configuration (**Figure 1b and Figure S2**) with our designed membrane electrode assemblies (MEA) for ECH of lignocellulosic bio-oil derived aromatics. The cathode layer, proton exchange membrane (PEM) and anode layer are closely packed to fabricate the MEA (**Figure S2b**) to shorten the distance between anode and cathode, and this minimizes the Ohmic resistance and cell potentials. A catalyst screening shows that Rh catalysts accelerate ECH of lignin monomer guaiacol and suppress the competing HER compared to other metal catalysts (e.g., Pt, Pd, Cu, Ni, Ir, and Ru) (**Figure S3**). The Rh nanoparticles of ~5 nm diameter (**Figure S4**) were coated on the carbon fiber substrate that is common electrolyte diffusion material (**Figure S5**). This Rh coated carbon fiber (Rh/CF) diffusion cathode is readily permeable to electrolytes and electrically conductive,³⁵ leading to a high mass transfer rate and a low full-cell voltage. Using this designed electrocatalytic system, we selectively upgraded three representative lignocellulose-derived aromatic monomers to high value chemicals at industrial-scale productivity level under ambient temperature and pressure.

(Page 2 in supporting information) **Electrocatalytic hydrogenation performance and product analysis.** The ECH of bio-oil compounds was performed using a two-chamber

flow-cell separated by a Nafion 117 membrane. Cathode and anode are closely against the Nafion membrane to minimize the distance between electrodes and the membrane (Figure S2b), and this lowers internal Ohmic resistance and accelerates ion transportation and associated ECH rates². For the control experiments in an H-cell, the electrolyte (30 mL) was magnetically stirred at 1000 RPM. The carbon felts loaded with Rh were restricted to a geometric size of 1 cm x 1 cm as work electrodes with thickness of 3 mm (cathode). 50 mL of 0.2 M HClO₄ containing a specific concentration of selected bio-oil compound (e.g., 50 mM FA, 80 mM syringol, or 120 mM guaiacol) was used as the catholyte. For ECH of FA, catholyte also contained 30% of ethanol and was purged with Argon gas to evacuate the air before ECH. The electrolyte was circulated through the flow cell at 150 ml/min using peristaltic pumps (Lead Fluid BT100S-1). IrO₂ on titanium felt and 0.2 M HClO₄ solution were used as anode and anolyte, respectively. Cathodic potentials were measured against an Ag/AgCl reference electrode (saturated KCl). Full-cell voltages were measured against anode.

Figure S2. (a) The electrolyte was circulated through the flow cell. (b) Schematic

illustrating our custom-made membrane electrode assemblies (MEA) consisting of a catalyst coated diffusion layer, proton exchange membrane (PEM) and IrO₂/Ti mesh anode. These three layers are closely packed to minimize the Ohmic resistance and electrode potentials.

REVIEWER COMMENTS

Reviewer #1 (Remarks to the Author):

Suggestions for the revised MS:

1. Please list the aqueous electrolyte solubilities of furan and all the lignin monomers investigated in this work.
2. In the revised MS the authors indicate that a highly-permeable gas-diffusion electrode is needed for effective performance. The question is than why to use gas-diffusion electrode at all? As the name implies a gas-diffusion electrode is facilitating the gas transport to the reaction sites. Why not use a porous hydrophilic electrode? Please explain.

Reviewer #2 (Remarks to the Author):

The authors effectively addressed this reviewer's comments.

Reviewer #3 (Remarks to the Author):

It can be published.

Additionally, the Editor asked me to comment on the authors responses to Reviewer #4. These comments are below.

1) I think that Reviewer #4's concerns are reasonable.

In this study, the authors explored the potential advantages of an electrocatalytic hydrogenation system for lignocellulosic bio-oil-derived aromatics. The system comprised a highly-electrolyte-permeable Rh diffusion cathode with membrane electrode assemblies (MEA) in a flow-cell configuration mode. The authors highlighted that the close-packing structure of MEA offers benefits in minimizing Ohmic resistance and cell potentials.

However, it's worth noting that the concepts of close-packed MEA, flow-cell configuration, and high electrolyte permeable cathodes are well-established within the electrochemistry research community. Therefore, these aspects may not be novel to experienced researchers in the field.

One concern raised in the study is the use of expensive noble metals such as Rh and Ir, which are considerably costlier than Pt, especially when utilized in nanoparticle form with low atomic utilization efficiency. Furthermore, the instability of IrO₂ under acidic oxygen evolution conditions may raise questions about the observed high stability of the electrocatalytic hydrogenation (ECH) system, particularly at elevated potentials and current densities. Due to these factors, the claims presented in

the study might not be as appealing or impactful to experienced researchers in the field.

Furthermore, it's imperative to address concerns about the performance comparison of the studied electrode with previously developed systems. The accuracy of such comparisons depends on the appropriate and standardized determination of electrode surface area. Without proper methodology and measurements, the claimed several times enhancement compared to other materials lacks a meaningful basis for comparison (Please find more detailed discussions below).

The primary novelty of this research lies in the record-high performance achieved by the designed ECH system. This paper is more about engineering rather than material science. Therefore, it is crucial for the authors to provide a more in-depth focus on the novel engineering aspects of their ECH system.

Taking Comments #7 and #12 as examples. The authors briefly mentioned the flow-cell configuration with the designed membrane electrode assemblies (MEA) and provided a one-sentence description of the closely packed MEA. Typically, MEA in electrocatalytic experiments is also set up with a closely packed structure. Thus, it is essential to clarify what sets their closely-packed MEA apart from the typical MEA used in electrocatalytic experiments. The authors should clearly describe the fabrication method employed for their closely-packed MEA (if novel). Similarly, more comprehensive details are needed regarding the construction of the flow cell. A mere photograph and a simple scheme (Figure S2) are insufficient to provide a clear understanding of the unique design. The authors should offer a thorough description of the flow cell construction, highlighting the distinctive features that contribute to the exceptional performance of the ECH system. Additional figures or diagrams could aid in better visualizing the innovative aspects of the closely-packed MEA and the flow cell.

2) While the authors did present additional characterization results, it appears that the concerns raised by Reviewer #4 have not been adequately addressed. The analysis of the electrode characterization was not well-conducted, and the comparison remains biased. For example:

a. In response of comment #1:

- Using the geometric area for the current density comparison is inappropriate (please see more details in question 3). The cited reference (Nat. Nanotechnol. 16, 140-147 (2021)) does not mention about the use of the geometric area for current density determination. Inappropriate determination of surface area could cause erroneous exaggeration of the current density.
- The unit of Rh loading should be either mg/cm² or mg/g catalyst, not mg.
- The comparison of “The partial intrinsic current density is 0.365 mA/cm² to the target products (methoxy-cyclohexane) from ECH of guaiacol, representing ~3x increase compared to other guaiacol ECH reports. The TOFs are 1158 h⁻¹ and 1402 h⁻¹ on Rh/CF catalyst at applied current density of 300 and 400 mA/cm² respectively, representing a ~2x increase compared to the best prior report (J. Chem. Phys. 156, 104703 (2022)).” is not reasonable. Intrinsic current density and TOF are functions of potential applied, catalyst loading, and total current density. The performances measured in referenced literature were not measured at either the same applied potential or total current density. A comparison without any same basis is unreasonable.

- The Cu-UPD characterization was not well-conducted. Typically, the Cu should be deposited at a specific potential, and then only the stripping region of Cu should be scanned to ensure accurate results. This approach helps avoid potential interference from H-deposition, oxidation of Rh, or over-deposition of Cu. Additionally, in Figure S14, the integrated area should be annotated for better clarity and understanding. Besides, it is worth noting that the current density reported ($\sim 15 \text{ mA cm}^{-2}$) is significantly higher than typically measured ($1\text{-}0.1 \text{ mA cm}^{-2}$). This observation further raises concerns about the inappropriate use of the geometric area for determining the current density.
- The calculation of number of surface Rh atom (NRh) should be provided.

b. Others:

- The cyclic stability test (Figure S11) is strange. The authors should provide the experimental details.
- The XPS (Figure S13) should be fitting-analyzed to determine the fraction of Rh and Rh oxide in the materials.

3) The authors carried out BET and ECSA characterizations to assess the catalyst's surface area. However, significant concerns remain regarding the inappropriate use of the geometric surface area for determining current density. Typically, employing the geometric area for current density determination is only suitable for smooth planar electrodes with a very thin layer of coated electrocatalyst film (usually much smaller than 1 mm, forming quasi-2D electrodes). In contrast, the cathode used in this study is relatively thick (3 mm) and should be considered a 3D electrode. The actual area of this electrode is much larger than its geometric area.

Using the geometric surface area to compare the electrode's performance with other works where the electrode was prepared by drop-casting an electrocatalytic ink on the electrode surface is inappropriate. Such a comparison could lead to exaggerated and erroneous conclusions.

4) I find the authors' response to point 8 to be unsatisfactory. As previously mentioned, significant concerns arise from the comparison presented in Table S1. Typically, for a fair comparison, measurements should be conducted at the same potential or the same total current density. Additionally, the mass loading of the catalysts plays a crucial role in determining current density and should be considered.

I understand that the ECH applications may not be as common, and there might be a lack of universal testing protocols or standardized performance parameters. Thus, I suggest that the authors include Faradaic efficiency (FE) data at $J = 100 \text{ mA/cm}^2$. This additional data would enable a more reasonable comparison with previously reported materials, considering their performance at a similar current density (those near the end of Table S1).

Furthermore, to provide a comprehensive comparison, I recommend adding columns for potential applied and mass loading in Table S1. Such improvements will clearly highlight the advantages of the studied ECH system over existing materials and improve the novelty of the research.

5) As discussed above, I think that the comments of Reviewer #4 have not been well-addressed to support the publication. However, as commented before, this work is interesting. Thus, I recommend further revisions to make this work fully attain Nature Communications' high scientific standards.

Manuscript ID: NCOMMS-23-02088

Title: “*Electrocatalytic valorization of lignocellulose-derived aromatics at industrial-scale current densities*”

We appreciate the referees' suggestions that motivated us to improve the quality of this work, and have revised the manuscript carefully in the light of the referees' advice. Point-by-point responses to the reviewers' comments are provided. Please find hereafter our answers (in **blue**) to the reviewers' comments (in **black**), and for convenience, the changed parts, and newly added parts for the revised manuscript have been highlighted in a yellow background.

REVIEWER COMMENTS

Reviewer #1 (Remarks to the Author):

The present work discusses the electrochemical hydrogenation of lignin monomers and furan. The following recommendations are made for the revised manuscript:

Re: Thank you for the valuable recommendations.

1. Please list the aqueous electrolyte solubilities of furan and all the lignin monomers investigated in this work.

Re: We have now listed a table of all the lignin monomers as below.

Name	solubilities
Furfuryl alcohol	Miscible
Tetrahydrofurfuryl alcohol	Miscible
Guaiacol	17 g/L (15 °C) 0.1349 mol/L
Syringol	20 g/L (25 °C) 0.1299 mol/L
2-methoxycyclohexanol	61.8 g/L (25 °C) 0.475 mol/L
2-methoxycyclohexanone	17.2 g/L (25 °C) 0.134 mol/L
Cyclohexanol	40 g/L (25 °C) 0.3594 mol/L
Cyclohexanone	90 g/L (25 °C) 0.917 mol/L

2. In the revised MS the authors indicate that a highly-permeable gas-diffusion electrode is needed for effective performance. The question is than why to use gas-diffusion electrode at all? As the name implies a gas-diffusion electrode is facilitating

the gas transport to the reaction sites. Why not use a porous hydrophilic electrode? Please explain.

Re: The ECH electrode consists of Rh nanoparticles coated on the porous carbon fiber substrate that is electrolyte diffusion material. The originally hydrophobic CF is rendered hydrophilic through the Rh metal coating. In this study, we then employed the Rh/CF as electrolyte diffuse electrode.

Reviewer #2 (Remarks to the Author):

The authors effectively addressed this reviewer's comments.

Re: Thank you for your recognition.

Reviewer #3 (Remarks to the Author):

It can be published.

Re: We appreciate your support and recognition of this work.

Additionally, the Editor asked me to comment on the authors responses to Reviewer #4. These comments are below.

Re: We appreciate your suggestions and helpful comments. We have now revised manuscripts accordingly. Please see the responses below.

1. I think that Reviewer #4's concerns are reasonable.

In this study, the authors explored the potential advantages of an electrocatalytic hydrogenation system for lignocellulosic bio-oil-derived aromatics. The system comprised a highly-electrolyte-permeable Rh diffusion cathode with membrane electrode assemblies (MEA) in a flow-cell configuration mode. The authors highlighted that the close-packing structure of MEA offers benefits in minimizing Ohmic resistance and cell potentials.

However, it's worth noting that the concepts of close-packed MEA, flow-cell configuration, and high electrolyte permeable cathodes are well-established within the electrochemistry research community.

Therefore, these aspects may not be novel to experienced researchers in the field.

Re: While MEA configuration is well-established in the field of electrochemistry, particularly in applications such as fuel cells and water electrolyzers, their utilization in electrochemical organic synthesis remains limited. The conventional experimental setup of choice has typically been the H-cell. Previous studies conducted using H-cells often operated at current densities below those required for industrial-scale productivity.

In our study, we have integrated the MEA configuration into organic reaction systems and yielding gratifying research outcomes. Our custom-designed MEA-like flow cell features a cathode composed of a porous carbon felt coated with rhodium (acting as the electrolyte diffusion electrode). This porous electrode exhibits excellent permeability to electrolytes and high electrical conductivity, resulting in rapid mass

transport and reduced overall cell voltage. We applied this system to the upgrading of three representative bio-oil monomers, demonstrating its potential for enhancing the conversion of biomass-derived bio-oil.

2. One concern raised in the study is the use of expensive noble metals such as Rh and Ir, which are considerably costlier than Pt, especially when utilized in nanoparticle form with low atomic utilization efficiency. Furthermore, the instability of IrO₂ under acidic oxygen evolution conditions may raise questions about the observed high stability of the electrocatalytic hydrogenation (ECH) system, particularly at elevated potentials and current densities. Due to these factors, the claims presented in the study might not be as appealing or impactful to experienced researchers in the field.

Re: We did the stability tests with monitoring the potentials of IrO₂ anode and Rh/CF cathode, and found they were stable under elevated current densities. In addition, the good stability of IrO₂ under acidic oxygen evolution conditions has been well summarized in a recent Review paper about the stability of OER catalysts (*Angew. Chem. Int. Ed.* 2017, 56, 5994–6021).

Moreover, similar to our experimental conditions, the literature (*Nat. Commun.*, 2021, 12, 6007) confirmed the good stability of IrO₂ for acidic OER at elevated potentials and current densities (at 250 mA/cm_{geo}²).

Although noble metals were used, ECH system yielded high efficiency and industrial-scale productivity for the high-value products, leading to the lowered plant-gate levelized cost. This is supported by our TEA studies. Moreover, Rh can be recycled by calcination to remove carbon felt later, further reducing the costs.

3. Furthermore, it's imperative to address concerns about the performance comparison of the studied electrode with previously developed systems. The accuracy of such comparisons depends on the appropriate and standardized determination of electrode surface area. Without proper methodology and measurements, the claimed several times enhancement compared to other materials lacks a meaningful basis for comparison (Please find more detailed discussions below).

Re: We will discuss it in the follow-up question.

4. The primary novelty of this research lies in the record-high performance achieved by the designed ECH system. This paper is more about engineering rather than material science. Therefore, it is crucial for the authors to provide a more in-depth focus on the novel engineering aspects of their ECH system.

Re: Thank you for the suggestion, we will describe the corresponding details in the follow-up question.

5. Taking Comments #7 and #12 as examples. The authors briefly mentioned the flow-cell configuration with the designed membrane electrode assemblies (MEA) and provided a one-sentence description of the closely packed MEA. Typically, MEA in electrocatalytic experiments is also set up with a closely packed structure. Thus, it is essential to clarify what sets their closely-packed MEA apart from the typical MEA used in electrocatalytic experiments. The authors should clearly describe the fabrication method employed for their closely-packed MEA (if novel). Similarly, more comprehensive details are needed regarding the construction of the flow cell. A mere photograph and a simple scheme (Figure S2) are insufficient to provide a clear understanding of the unique design. The authors should offer a thorough description of the flow cell construction, highlighting the distinctive features that contribute to the exceptional performance of the ECH system. Additional figures or diagrams could aid in better visualizing the innovative aspects of the closely-packed MEA and the flow cell.

Re: We add a more comprehensive schematic diagram of the MEA flow cell with detailed structure and parameters (Figure S2c in Supplementary Information). The cathode layer, proton exchange membrane (PEM) and anode layer are closely packed to fabricate the MEA to shorten the distance between anode and cathode, and this minimizes the Ohmic resistance and cell potentials. The MEA flow-cell system shows a significantly lower full-cell voltage of ~ 3 V at 200 mA/cm^2 than does the H-cell system (~ 10 V) (Figure 2a in Manuscript).

Figure S2c. Scheme of MEA flow-cell with the detailed parameters.

The traditional methods for MEA preparation consist of catalyst coated substrate (CCS) and catalyst coated membrane (CCM) methods, showed in Figure 1 as below. For CCS method, the catalyst is directly coated onto gas diffusion layers (GDLs) to fabricate cathode GDL and anode GDL. These GDLs, with a catalyst layer, are pressed on both sides of the proton exchange membrane (PEM) using a heat-pressing technique to assemble the MEA. For CCM method, the catalyst is coated on both sides of the PEM. The cathode and anode GDLs are individually placed on both sides of the catalyst layers, followed by heat-pressing to assemble the MEA. Traditional GDLs often use carbon paper or carbon cloth, which typically has a porosity ~ 0.8 . We here used carbon felt as GDL, which has a higher porosity ~ 0.98 (*J. Mater. Chem.*, 1999, **9**, 419-426) and a better electrolyte permeability and mass transport. Carbon felt has been extensively applied in lithium-ion batteries and flow batteries. Our MEA preparation method is similar to the CCS method, where the catalyst (Rh) is coated onto the GDL (carbon felt), then the cathode GDL, PEM and anode are closely packed. In contrast to H-cell, using MEA had been seldomly reported yet in the field of electrochemical synthesis of organic compounds.

Figure 1 Two different MEA fabrication methods (*Chin. J. Chem. Eng.* 33, 2021, 1–16)

6. While the authors did present additional characterization results, it appears that the concerns raised by Reviewer #4 have not been adequately addressed. The analysis of the electrode characterization was not well-conducted, and the comparison remains biased. For example:

In response of comment #1:

6.1 Using the geometric area for the current density comparison is inappropriate (please see more details in question 3). The cited reference (*Nat. Nanotechnol.* 16, 140-147 (2021)) does not mention about the use of the geometric area for current density determination. Inappropriate determination of surface area could cause erroneous exaggeration of the current density.

Re: In the field of electrocatalysis, geometric area has been widely used for determination of current densities. Common porous materials like nickel foam typically have thicknesses exceeding 1mm, yet current densities were still calculated using geometric area, please refer to the literatures (*Adv. Mater.* 2023, 35, 2208284; *Nat. Commun.* 2022 13, 5297). Moreover, in Table 1 shown as below, these literatures clearly state using geometric area of different electrodes (thickness: 1 ~ 6 mm) including carbon felt for the calculation of current densities. Thus, we believe it is reasonable to use current density based on geometric area of our Rh/CF (3 mm thick) electrode to compare to literatures. Furthermore, we have given the current densities based on ECSA in manuscript.

In the last revision, we performed Cu underpotential deposition and obtained the

electrochemical surface area (ECSA: 314 cm² for 6.2 mg Rh) of our Rh/CF electrode. Then we calculated the specific current density and turnover frequency (TOF) using the ECSA instead of geometric area, which was added in the latest MS submitted. To be clear, ECH performance comparisons to literature in Table S1 in Supplementary Information, the listed current densities of our results and literature are all based on geometric areas. Using ECSA instead of geometric area, our calculated specific current density (e.g., at total current density of 300 mA/cm²) is still superior to those specific current densities in previous literatures. And the TOF of our Rh/CF catalyst is ~ 3x compared to the best value in prior reports, which clearly demonstrates the superior catalytic capability of our ECH system.

Table 1.

Title	Thickness	Geometric area used for current density	References
Simultaneous Oxidative Cleavage of Lignin and Reduction of Furfural via Efficient Electrocatalysis by P-Doped CoMoO ₄	1mm	1 cm ²	Adv. Mater. 2023, 35, 2208284
Electrocatalytic hydrogenation of quinolines with water over a fluorine-modified cobalt catalyst	1mm	1 cm ²	Nature Communications (2022) 13:5297
Amine Coordinated Electron-Rich Palladium Nanoparticles for Electrochemical Hydrogenation of Benzaldehyde	6mm	2 cm ²	Adv. Funct. Mater. 1 cm ² 2023, 33, 2214588
Three-Dimensional PbO ₂ -Modified Carbon Felt Electrode for Efficient Electrocatalytic Oxidation of Phenol Characterized with In Situ ATR-	1.5mm	4 cm ²	J. Phys. Chem. C 2022, 126, 2, 912–921

FTIR			
Stabilities, Regeneration Pathways, and Electrocatalytic Properties of Nitroxyl Radicals for the Electrochemical Oxidation of 5-Hydroxymethylfurfural	3mm	1 cm ²	ACS Sustainable Chem. Eng. 2019, 7, 13, 11138–11149
Ternary Alloys Enable Efficient Production of Methoxylated Chemicals via Selective Electrocatalytic Hydrogenation of Lignin Monomers	3mm	1 cm ²	J. Am. Chem. Soc. 2021, 143, 41, 17226–17235

6.2 The unit of Rh loading should be either mg/cm² or mg/g catalyst, not mg.

Re: Thank you. We have adjusted the units in the article, converting them to mg/cm² as follows:

The Rh loading is about 6.2 mg/cm².

6.3 The comparison of “The partial intrinsic current density is 0.365 mA/cm² to the target products (methoxy-cyclohexane) from ECH of guaiacol, representing ~3x increase compared to other guaiacol ECH reports. The TOFs are 1158 h⁻¹ and 1402 h⁻¹ on Rh/CF catalyst at applied current density of 300 and 400 mA/cm² respectively, representing a ~2x increase compared to the best prior report (J. Chem. Phys. 156, 104703 (2022)).” is not reasonable. Intrinsic current density and TOF are functions of potential applied, catalyst loading, and total current density. The performances measured in referenced literature were not measured at either the same applied potential or total current density. A comparison without any same basis is unreasonable.

Re: We now add the turnover frequency (TOF) derivation process in supplementary information as below:

$$\text{TOF} = \frac{N_p}{N_{\text{Rh}} \times t}$$

$$J = \frac{I}{A}$$

$$\text{FE} = \frac{N_p \cdot Z_p \cdot F}{I \cdot t} \times 100\%$$

$$J_p = J \cdot \text{FE}$$

$$\text{so, } \frac{N_p}{t} = \text{FE} \cdot \frac{I}{Z_p \cdot F} = \text{FE} \cdot \frac{J \cdot A}{Z_p \cdot F} = \frac{J_p \cdot A}{Z_p \cdot F}$$

$$\text{TOF} = \frac{N_p}{N_{\text{Rh}} \cdot t} = \frac{J_p \cdot A}{N_{\text{Rh}} \cdot Z_p \cdot F} = \frac{A}{Z_p \cdot F} \cdot \frac{J_p}{N_{\text{Rh}}}$$

Where N_p is the moles of total target products formed; N_{Rh} is the moles of surface Rh atoms; t is the time duration for production of N_p moles target products; F is the faradaic constant; J is total current density; I is total current; A is geometric area of work electrode (1 cm^2 for Rh electrode); Z_p is the number of electrons transferred for one molecule of target product formed;

From the formula, it is evident that TOF is directly proportional to the partial current density (J_p), which is FE multiply total current density (J). So, the partial current density to target product directly reflects the intrinsic catalytic activity (TOF) of ECH system. Large partial current density (J_p) certainly requires large total current density (J), and large total current density requires high potentials (here more negative for ECH reactions).

As we know, high potentials lead to stronger competing side reactions (e.g., HER), and cause FE to decrease. So, continuous increasing the total current density cannot always enhance partial current density due to FE will drop too much at high potentials. In order to maximize the partial current density, we engineered ECH catalysts and device to unite the total current density and FE, which demonstrated the superior intrinsic catalytic activity for our Rh/CF diffusion electrode and ECH system. If we compare our TOF with literatures at the same total current densities (i.e., much lower than ours), it will surely lower down our TOF value artificially, because partial current

density (J_p) is lowered down. But this comparison cannot show the real catalytic capability of our Rh/CF electrode, in terms of the limitation of TOF. Thus, we compared our maximum TOF value at the optimized partial current density to the literatures' maximum TOF values, which is fair and reasonable. Moreover, TOF is inversely proportional to the amount of active surface Rh atoms (N_{Rh}) that is related to the catalyst loading mentioned by you. But, TOF has been already calculated with the normalization to N_{Rh} , so comparison of TOF values based on different catalyst loading is fair as well. We have added the catalyst loading column in Table S1 in supplementary information.

The partial intrinsic current density has the same catalytic sense with partial current density (J_p) as discussed above, just replacing the geometric area with ECSA for calculating current densities. We compared geometric area (or ECSA) derived partial current densities (or partial intrinsic current density) to literatures in the same calculation way, i.e., geometric area (or ECSA). Concerning why our partial current densities are superior to literatures, as discussed in the last paragraph, the up-limitation of partial current density is directly related to and determined by the intrinsic catalytic capability of the catalyst and ECH system.

6.4 The Cu-UPD characterization was not well-conducted. Typically, the Cu should be deposited at a specific potential, and then only the stripping region of Cu should be scanned to ensure accurate results. This approach helps avoid potential interference from H-deposition, oxidation of Rh, or over-deposition of Cu. Additionally, in Figure S14, the integrated area should be annotated for better clarity and understanding. Besides, it is worth noting that the current density reported ($\sim 15 \text{ mA cm}^{-2}$) is significantly higher than typically measured ($1\text{-}0.1 \text{ mA cm}^{-2}$). This observation further raises concerns about the inappropriate use of the geometric area for determining the current density.

Re: We appreciate the valuable suggestions. We conducted the Cu-UPD experiment again, and updated Figure S14 with a clearer annotation as below. We have confirmed the potential window ($0.103 \sim 0.503$ vs. Ag/AgCl or $0.3 \sim 0.7$ V vs. RHE) that we applied only allows under potential deposition of Cu, avoiding H-desorption, oxidation

of Rh and over-deposition of Cu. Figure S14 shows the current density peak at ~ 10 mA/cm², a little bit higher than ~ 6 mA/cm² of Cu-UPD on PtRu reported by literature (*J. Phys. Chem. B* 2002, 106, 5, 1036–1047). Our Rh/CF catalyst has a relatively high loading of 6.2 mg/cm² and ECSA (314 cm² for 6.2 mg Rh), leading to a higher Cu-UPD current density. For example, in literature (*Chem Electro Chem* 2019, 6, 1990–1995), the loading is 6.6 μ g/cm², and the ECSA is 9.24 cm², which explains the lower current density (~ 0.2 mA/cm²) reported in their experiments.

Figure S14. CV diagram for Cu_{UPD} of Cu on Rh/CF cathode.

We also conducted hydrogen underpotential deposition (H_{UPD}) test, and obtained an ECSA of 5.23 m²/g, in good agreement with the value of 5.073 m²/g derived from Cu_{UPD}. We have added it in Supplementary Information as Figure S15.

Figure S15. CV diagram for H_{UPD} of Rh/CF cathode.

6.5 The calculation of number of surface Rh atom (N_{Rh}) should be provided.

Re: Due to Cu deposition by UPD method as a monolayer on the Rh surface, the quantity of deposited Cu (N_{Cu}) is equal to that of surface Rh atom. The amount of deposited Cu can be calculated from the shaded stripping peak area in Figure S14 using the following formula:

$$Q_{Cu} = \frac{I * V}{\text{scan rate}}$$

$$Cu^{2+} + 2e^{-} \rightarrow Cu_{UPD}$$

$$N_{Rh} = N_{Cu} = Q_{Cu}/2F$$

N_{Rh} is the moles of surface Rh atoms, Q_{Cu} is the measured integral charge, F is the faradaic constant.

7. Others:

7.1 The cyclic stability test (Figure S11) is strange. The authors should provide the experimental details.

Re: We changed the catholyte (1L of 0.2 M $HClO_4$ containing 120 mM guaiacol) every 12 hours to ensure enough reactants for ECH reaction, constituting a single cycle. Faradaic efficiency for each cycle was assessed. Throughout this stability test, we changed catholyte for 5 times, resulting in 5 cycles a cumulative duration of 60 hours. We have now added the experimental details to Figure S11 caption.

7.2 The XPS (Figure S13) should be fitting-analyzed to determine the fraction of Rh and Rh oxide in the materials.

Re: We remeasured XPS of Rh catalyst with higher resolution on PHI Genesis 900 and added the fitting of Rh 3d in Supplementary Information (Figure S13), where the fraction of Rh oxide is ~3 at%. A small amount of surface oxide induced by air oxidation is inevitable, which is a common phenomenon in related researches (*Carbon Energy* 2022, 4, 283-293). We removed the surface oxides by running HER before ECH experiments, as described in Supplementary Information previously:

“The cathode was further reduced in 0.1 M Na₂SO₄ solution by applying a potential of -1 V vs. an Ag/AgCl reference electrode (saturated KCl) for 5 minutes.”

Figure S13. (a) XPS of Rh catalysts before and after 32-h reaction, showing no apparent change of surface chemical composition after ECH. (b) Fitting of Rh 3d before reaction, showing a small amount of Rh oxide (fitted spectra in blue) due to air oxidation.

7.3 The authors carried out BET and ECSA characterizations to assess the catalyst's surface area. However, significant concerns remain regarding the inappropriate use of the geometric surface area for determining current density. Typically, employing the geometric area for current density determination is only suitable for smooth planar electrodes with a very thin layer of coated electrocatalyst film (usually much smaller

than 1 mm, forming quasi-2D electrodes). In contrast, the cathode used in this study is relatively thick (3 mm) and should be considered a 3D electrode. The actual area of this electrode is much larger than its geometric area.

Using the geometric surface area to compare the electrode's performance with other works where the electrode was prepared by drop-casting an electrocatalytic ink on the electrode surface is inappropriate. Such a comparison could lead to exaggerated and erroneous conclusions.

Re: In electrochemical experiments, thick substrates are used as electrodes, and current density is still usually calculated based on geometric area. Common porous materials like nickel foam typically have thicknesses exceeding 1mm, yet current densities were still calculated using geometric area, please see the literatures (*Adv. Mater.* 2023, 35, 2208284; *Nat. Commun.* 2022 13, 5297). We compared geometric area (or ECSA) derived current densities to literatures in the same calculation way, i.e., geometric area (or ECSA), ensuring a fair comparison. Please refer to our answers to your Question 6.1.

7.4 I find the authors' response to point 8 to be unsatisfactory. As previously mentioned, significant concerns arise from the comparison presented in Table S1. Typically, for a fair comparison, measurements should be conducted at the same potential or the same total current density. Additionally, the mass loading of the catalysts plays a crucial role in determining current density and should be considered.

Re: We have added the catalyst loading column in Table S1 in supplementary information. Concerning comparisons in Table S1, we think it is fair and have given sufficient explanations in the replies to your Question 6.3.

One of the significant strengths of this study lies in its application of industrial-level current density. It is well-known that the commercialization of electrocatalysis is often hindered by the limitation of lower current densities, which compromises the feasibility of industrial implementation. In industrial production, higher current densities often lead to an increase in unwanted side reactions and a reduction in the faradaic efficiency of the target product. However, the MEA ECH system employed in

this study, exhibited a high faradaic efficiency and stability under industrial relevant current conditions, a distinguishing advantage of this work.

7.5. I understand that the ECH applications may not be as common, and there might be a lack of universal testing protocols or standardized performance parameters. Thus, I suggest that the authors include Faradaic efficiency (FE) data at $J = 100$ mA/cm². This additional data would enable a more reasonable comparison with previously reported materials, considering their performance at a similar current density (those near the end of Table S1).

Re: I agree to you about there is lacking of universal testing protocols or standardized performance parameters to evaluate ECH. We have added relevant data as your suggestions in Table S1, and please find it as below.

Table S1. Selected prior reports for electrocatalytic biorefinery of bio-oil aromatics. Abbreviations: Current density (J); faradaic efficiency (FE); partial current density (J_p); KA oil is the mixture of cyclohexanone ketone and cyclohexanol alcohol.

System	Catalyst	T (°C)	Bio-oil compound	Product	J mA/cm ²	FE %	J_p mA/cm ²	Loading mg/cm ²	Potential vs. RHE (V)	Reference
MEA	Rh/CF	25	Guaiacol	Methoxycyclohexane	100	68	68	6.2	-0.10	This work
MEA	Rh/CF	25	Furfural alcohol	Tetrahydrofurfuryl alcohol	200	67	134	6.2	-0.30	This work
MEA	Rh/CF	25	Guaiacol	Methoxycyclohexane	300	64	192	6.2	-0.58	This work
Improved H-cell	Rh/CF powders	18	Methoxyphenol	4-Methoxycyclohexanone	21	35	7.35	0.2	-0.4	J. Catal. , 2016, 344 263-272.

mixture										
Flow cell	PtNiB	60	Guaiacol	KA oils	5	80 ~ 90	4 ~ 4.5	2	-0.21	Adv. Funct. Mater. 2019, 29, 1807651. ⁶
H-cell	Pt/C	50	Furfural	Furfural alcohol	30	85	25.5	1	-0.25	Electrochim. Acta. 2014, 135 139–146. ⁷
PEM fuel cell	Pd/C	25	Furfural	Tetrahydrofurfuryl alcohol	19	25	4.8	1	-0.22	Green Chem. 2013, 15, 1869-1879. ⁸
H-cell	Raney Ni	75	Mequinol	4-Methoxycyclohexanol	8	26	2.1	12		Green Chem. 2015, 17, 601-609. ⁹
H-cell	Ru	50	Guaiacol	Methoxycyclohexanes	25	19	4.8			Green Chem. 2012, 14, 2540-2549. ¹⁰
H-cell	Ru	80	Syringol	Methoxycyclohexanes	25	49	12.3			Green Chem. 2012, 14, 2540-2549. ¹⁰
H-cell	Ru/ACC	80	Guaiacol	2-methoxycyclohexanol	100	30	30			Green Chem. , 2012, 14, 2540–2549. ¹⁰
H-cell	Ru	80	Guaiacol	Cyclohexanol	22	45	9.9			ACS Sustain. Chem. Eng. 2019, 7, 8375-8386. ¹¹
H-cell+ slurry reactor	Pt/C	50	Guaiacol	2-methoxycyclohexanone	109	72	78.5	9.5	-1.7	J. Appl. Electrochem. 2021, 5151-63. ¹²
H-cell+ slurry reactor	Pt/C	60	Guaiacol	Cyclohexanone	146	35	51	2.55	-2.1	Green Chem. 2022, 24, 7469-7480. ¹³
H-cell+ slurry reactor	Pt/C	50	Guaiacol	2-methoxycyclohexanol	109	94	102	2.4	-1.72	ChemSusChem 2020, 13, 629-639. ¹⁴
H-cell+ slurry reactor	Pt/C	24	Guaiacol	Cyclohexanol	150	75	112.5	1.8	-1.1	ACS Sustainable Chem. Eng. 2021, 9, 13164-13175. ¹⁵
Flow cell	Pd	25	Benzaldehyde	Benzyl alcohol	15	52	7.8	0.25	-0.49	ACS Sustain. Chem. Eng. 2020, 8, 4407-4418. ¹⁶

7.6 Furthermore, to provide a comprehensive comparison, I recommend adding columns for potential applied and mass loading in Table S1. Such improvements will clearly highlight the advantages of the studied ECH system over existing materials and improve the novelty of the research.

Re: Thank you for this suggestion. We have added relevant data in Table S1.

8. As discussed above, I think that the comments of Reviewer #4 have not been well-addressed to support the publication. However, as commented before, this work is interesting. Thus, I recommend further revisions to make this work fully attain Nature Communications' high scientific standards.

Re: We appreciate your support and recognition of this work. We have conducted further revisions to the manuscript and provided comprehensive responses to the raised comments. With this effort, we hope to meet the standards of *Nature Communications*.

REVIEWERS' COMMENTS

Reviewer #1 (Remarks to the Author):

The authors made significant changes to the manuscript for improvement. However, this emphasis on high current density like it would be a car race, is unwarranted. There are many other factors that come into play for commercialization of the electrochemical process including the energy consumption etc. The fact that they worked at 300 mA/cm² is not a 'record' of any sorts, because the current efficiency was 64%. This paper can be accepted if the authors remove these unjustified references to 'records'.

Reviewer #3 (Remarks to the Author):

It can be published.

Manuscript ID: NCOMMS-23-02088

Title: “*Electrocatalytic valorization of lignocellulose-derived aromatics at industrial-scale current densities*”

We appreciate the referees' suggestions that motivated us to improve the quality of this work, and have revised the manuscript carefully in the light of the referees' advice. Point-by-point responses to the reviewers' comments are provided. Please find hereafter our answers (in **blue**) to the reviewers' comments (in **black**), and for convenience, the changed parts, and newly added parts for the revised manuscript have been highlighted in a yellow background.

REVIEWER COMMENTS

Reviewer #1 (Remarks to the Author):

The authors made significant changes to the manuscript for improvement. However, this emphasis on high current density like it would be a car race, is unwarranted. There are many other factors that come into play for commercialization of the electrochemical process including the energy consumption etc. The fact that they worked at 300 mA/cm^2 is not a 'record' of any sorts, because the current efficiency was 64%. This paper can be accepted if the authors remove these unjustified references to 'records'.

Re: We appreciate your recognition of this work. We claimed that the obtained “partial current density” is the record high up to now, which has considered the current efficiency of 64%. And, we have removed the “record” from the manuscript anyway, and it now reads as follows:

We achieved high faradaic efficiencies up to 64% at industrial-scale current densities of $300\text{--}500 \text{ mA cm}^{-2}$, representing high productivities (or partial current densities) to target products.

Reviewer #3 (Remarks to the Author):

It can be published.

Re: We appreciate your support and recognition of this work.